# ARE LARGE LANGUAGE MODELS GOOD TEMPORAL GRAPH LEARNERS?

## ABSTRACT

Large Language Models (LLMs) have recently driven significant advancements in Natural Language Processing and various other applications. While a broad range of literature has explored the graph-reasoning capabilities of LLMs, including their use as predictors on graphs, the application of LLMs to real-world evolving networks (or temporal graphs), remains relatively unexplored. Recent work studies synthetic temporal graphs generated by random graph models, but applying LLMs to real-world temporal graphs remains an open question. To address this gap, we introduce Temporal Graph Talker (TGTalker), a novel temporal graph learning framework designed for LLMs. TGTalker utilizes the recency bias in temporal graphs to extract relevant structural information, converted to natural language for LLMs, while leveraging temporal neighbors as additional information for prediction. TGTalker demonstrates competitive link prediction capabilities compared to existing Temporal Graph Neural Network (TGNN) models. Across five real-world networks, TGTalker performs competitively with state-of-the-art temporal graph methods while consistently outperforming popular models such as TGN and HTGN. Furthermore, TGTalker generates textual explanations for each prediction, thus opening up exciting new directions in explainability and interpretability for temporal link prediction.

## 1 INTRODUCTION

The field of artificial intelligence has witnessed remarkable progress through the development of Large Language Models (LLMs), beginning with the foundational Transformer architecture (Vaswani et al., 2017) and evolving through significant milestones including BERT (Devlin et al., 2019), GPT-3 (Brown et al., 2020), and ChatGPT (OpenAI, 2023). These models have demonstrated unprecedented capabilities across diverse domains, showcasing their versatility and potential for real-world applications. In Computer Vision, models such as LLaVA (Liu et al., 2023) and GPT-4 (OpenAI et al., 2023) have achieved remarkable multimodal understanding, while in Healthcare, Med-PaLM (Singhal et al., 2023b) and Med-PaLM 2 (Singhal et al., 2023a) have demonstrated expert-level performance on complex medical reasoning tasks. Beyond these domains, LLMs have shown promising results in scientific research (Schulz et al., 2025), legal analysis (Calzolari et al., 2024), and creative tasks (Franceschelli et al., 2025), highlighting their transformative potential across professional and academic fields.

A key advancement in LLM capabilities is the emergence of In-Context Learning (ICL), which enables models to adapt to new tasks without explicit fine-tuning (Dong et al., 2023; Coda-Forno et al., 2023; Sia et al., 2024; Brown et al., 2020). This paradigm encompasses three fundamental approaches: zero-shot learning, where models solve tasks directly from natural language instructions; one-shot learning, which provides a single example alongside instructions; and few-shot learning, which leverages multiple examples to demonstrate task patterns. This flexibility in learning approaches has opened new avenues for applying LLMs to complex structured data. To harness these capabilities for graph-structured data, recent research has explored various approaches to integrate LLMs with graph reasoning tasks. This integration is particularly relevant given that many real-world systems—from molecular structures (Beaini et al., 2024) and social networks (Hamilton et al., 2017) to transportation systems (Zheng & Bashir, 2022) and web data (Broder et al., 2000)—are naturally represented as graphs. Prior work has investigated methods for encoding graph structure for LLMs (Fatemi et al.,

Figure 1: Overview of TGTalker framework. TGTalker utilizes pre-trained LLMs to generate link prediction answers and explanations for real-world temporal graphs.

2024), combining graph encoders with LLMs (Fatemi et al., 2023), and fine-tuning LLMs specifically for graph tasks (Ye et al., 2023).

However, a significant gap exists in applying LLMs to temporal graphs (TGs), which represent evolving networks such as the Internet and social media platforms where entities and relations change over time. While the pioneering work of LLM4DyG (Zhang et al., 2024b) has demonstrated preliminary spatial-temporal understanding capabilities of LLMs on synthetic temporal graphs, these insights are limited to small-scale graphs with approximately 20 nodes. The potential of LLMs for real-world temporal graphs, which often contain thousands or millions of edges and exhibit complex temporal dynamics, remains largely unexplored.

To bridge this gap, we introduce Temporal Graph Talker (TGTalker), the first comprehensive framework that leverages LLMs for predictions on real-world temporal graphs. TGTalker employs a pre-trained LLM as its foundation while intelligently translating temporal graph structures into natural language representations tailored to specific tasks. In addition, LLMs naturally interfaces with users in the text space which can provide textual explanability unseen in existing Temporal Graph Neural Network (TGNN) approaches. Figure 1 shows an overview of the TGTalker framework where LLMs are applied for prediction and explanation on temporal graphs. TGTalker consists of four key components: (1) a *background set* that captures temporal context by incorporating relevant temporal links near the target link, (2) an *example set* that provides in-context learning samples through carefully selected question-answer pairs, (3) a *query set* that formulates prediction tasks in natural language, and (4) a *temporal neighbor sampling* mechanism that ensures efficient and relevant context selection. Our framework aligns with standard temporal link prediction tasks in temporal graph learning (Rossi et al., 2020; Yu et al., 2023; Huang et al., 2023b; Gastinger et al., 2024) while introducing novel capabilities for explainability and adaptability through LLM-based reasoning. Our main contributions are as follows:

- **LLM for real-world temporal graphs**. We present TGTalker, a novel framework that leverages LLMs for prediction on real-world temporal graphs. Our approach combines the recency bias inherent in temporal graphs with temporal neighbor sampling to convert the most relevant temporal graph structures to textual representations for LLM processing.
- **Strong link prediction performance.** Through extensive empirical evaluation across three open-source LLM families and five diverse real-world temporal graphs, we demonstrate that TGTalker achieves competitive performance with state-of-the-art TGNN. Notably, TGTalker consistently outperforms both the widely-used TGN architecture and three snapshot-based models across all evaluated datasets.
- **Temporal link explanations**. TGTalker introduces a novel paradigm of temporal link explanation by leveraging the natural language capabilities of LLMs. This enables the generation of human-readable explanations for model predictions, opening new directions for explainable TG methods.
- **Diverse explanation categories**. From LLM-generated explanations, we extract ten diverse explanation categories. Interestingly, some explanation categories are similar to existing TG algorithms that capture recent interactions or global popular destinations. These categories shed light on the diverse link pattern reasoning capability of LLMs.

## 2 RELATED WORK

**Temporal Graph Learning.** Temporal Graph Learning (TGL) focuses on modeling spatial and temporal dependencies in evolving networks. Following the taxonomy in (Kazemi et al., 2020),

methods are typically classified as either Continuous-Time Dynamic Graphs (CTDGs) or Discrete-Time Dynamic Graphs (DTDGs). CTDG methods (Longa et al., 2023), such as TGN (Rossi et al., 2020), EdgeBank (Poursafaei et al., 2022), TNCN (Zhang et al., 2024a), and GraphMixer (Cong et al., 2022), operate on irregular event streams, capturing fine-grained node interactions important for domains like social networks (Kumar et al., 2019) and financial transactions (Shamsi et al., 2024). In contrast, DTDG methods process graph snapshots at fixed intervals, combining GNNs with recurrent models (Yang et al., 2021; Kipf & Welling, 2016; Chen et al., 2022). While DTDGs offer computational efficiency by processing full snapshots and retaining historical context, they may suffer from reduced temporal fidelity (Huang et al.). The UTG framework (Huang et al.) addresses this gap by adapting snapshot-based models to event-based datasets. In our experiments, TGTalker is evaluated against representative CTDG and DTDG methods.

**In-Context Learning on Graphs.** Recent efforts have explored applying in-context learning (ICL) to graph-structured data, where challenges arise from translating complex relational information into linear prompts suitable for LLMs. Unlike conventional NLP tasks, effective graph ICL requires careful encoding of substructures and relational patterns. Frameworks such as PRODIGY (Huang et al., 2023a) propose pretraining objectives and model architectures tailored for graph-based pro3mpting. Other work investigates the innate ability of LLMs to infer patterns from structured examples with minimal adaptation (Li et al., 2025). A key challenge addressed by TGTalker lies in how to select, verbalize, and sample relevant temporal graph substructures to maximize ICL effectiveness.

**Graph Reasoning with Large Language Models.** Beyond in-context learning, a growing body of research examines the capacity of LLMs for more general graph reasoning tasks (Fatemi et al., 2024; Sanford et al., 2024; Behrouz et al., 2024; Perozzi et al., 2024). While LLMs demonstrate preliminary spatial and temporal understanding (Zhang et al., 2024b), they often struggle with dynamic graphs and tasks requiring complex multi-hop reasoning (Dai et al., 2024; Nguyen et al., 2024). Moreover, shortcut learning behaviors—where models produce correct outputs via flawed reasoning—have been widely observed. Nonetheless, the ability of LLMs to reason without task-specific fine-tuning positions them as attractive alternatives to specialized temporal graph neural networks (TGNNs). TGTalker builds on these insights by proposing methods to structure temporal information in ways that enhance LLMs' robustness and reasoning depth.

**LLMs for Temporal Knowledge Graphs.** Recent research adapts LLMs to Temporal Knowledge Graphs (TKGs), leveraging either ICL (Lee et al., 2023), semantic knowledge of temporal dynamics (Ding et al., 2024; Xia et al., 2024; Wang et al., 2024), or fine-tuning strategies (Liao et al., 2024). However, most TKGs encode rich textual descriptions for nodes and edges, allowing LLMs to rely on surface-level semantics rather than deeper structural reasoning. In contrast, TGTalker focuses on general temporal graphs where entities are represented abstractly (e.g., by identifiers), requiring models to reason over raw temporal-structural patterns without external semantic clues. This is crucial for evaluating the true relational reasoning abilities of LLMs at fine temporal granularities.

## 3 PRELIMINARY ON TEMPORAL GRAPH LEARNING

Following the formulation of Continuous Time Dynamic Graphs (CTDGs) in (Huang et al., 2023b; Rossi et al., 2020; Poursafaei et al., 2022; Luo & Li, 2022), we represent temporal graphs as timestamped streams of edges and formally defined as:

**Definition 1** (Temporal Graph). A temporal graph $\mathcal{G}$ can be represented as $\mathcal{G} = \{(s_1, d_1, t_1), (s_2, d_2, t_2), \ldots, (s_T, d_T, T)\}$, where the timestamps are ordered ($0 \leq t_1 \leq t_2 \leq \ldots \leq T$) and $s_i$, $d_i$, $t_i$ denote the source node, destination node and timestamp of the $i$-th edge.

Note that $\mathcal{G}$ can be directed or undirected, and weighted or unweighted. $\mathcal{G}_t$ is the cumulative graph constructed from all edges in the stream before time $t$ with nodes $\mathbf{V}_t$ and edges $\mathbf{E}_t$. We consider a fixed chronological split to form the training, validation and test set.

**Definition 2** ($k$-hop neighborhood in a temporal graph). Given a timestamp of interest $t$, denote $\mathcal{G}_t$ as the cumulative graph constructed by all temporal links before $t$. Given a node $u$, the $k$-hop neighborhood of $u$ before time $t$ is denoted by $\mathcal{N}_u^{t,k}$ and it is defined as the set of all nodes $v$ where there exists at least a single walk of length $k$ from $u$ to $v$ in $\mathcal{G}_t$.

In this work, we follow the standard *streaming setting* (Rossi et al., 2020) for evaluation in Continuous Time Dynamic Graphs (CTDGs). In this setting, the test set information is available for updating the

model representation (i.e. via a memory module or via temporal neighbor sampling); however, no gradient or weight updates are allowed at test time. In this way, the models can adapt to incoming data without expensive gradient updates.

## 4 METHODOLOGY

In this section, we present our Temporal Graph Talker (TGTalker) framework for applying Large Language Models on real-world temporal graphs. We start by discussing how to encode the temporal network structure into discrete text tokens for interfacing with LLMs.

### 4.1 APPLYING LLMS ON TEMPORAL GRAPH

**LLM Notation.** Consider a pre-trained Large Language Model (LLM) as an interface function $f$ where discrete tokens are received as input and generated as output via the token space $\mathbf{W}$, i.e. $f : \mathbf{W} \to \mathbf{W}$. Therefore, many tasks with LLM can be formulated as question and answer pairs, i.e., $\mathbf{A} = f(\mathbf{Q})$ where $\mathbf{Q} \in \mathbf{W}$ is the question or query and $\mathbf{A} \in \mathbf{W}$ is the answer of interest.

**Temporal Link Prediction with LLMs.** Given a temporal graph observed until time $t$, i.e. $\mathcal{G}_t$, the goal of temporal link prediction is to predict the destination node $d$ of a given source node $s$ at a future timestamp $t'$. Therefore, we can formulate temporal link prediction as a question-and-answer task for LLMs as follows:

$$\mathbf{A} = f(g(\mathcal{G}_t), q(\mathbf{Q}_{s,?,t'})) \tag{1}$$

where $\mathbf{Q}_{s,?,t'}$ is the link query, $g : \mathcal{G} \to \mathbf{W}$ is a temporal graph encoding function and $q : \mathbf{W} \to \mathbf{W}$ is the query encoding function. Thus, it is important to identify the appropriate temporal graph encoding function $g$ and query encoding function function $q$ such that the LLMs can provide the correct answer $\mathbf{A}$ while respecting the constraints from the LLM architecture such as maximum input size $l_{\max}$. Note that in practice, standard temporal graph learning methods treat the temporal link predicting problem as a ranking problem (Huang et al., 2023b; You et al., 2022) where the model outputs a probability for each considered node pair and then selects the node pair with the highest probability. With LLMs, it is possible to directly ask the model which destination node is the most probable and avoid the need to predict probability for all node pairs for ranking. Therefore, in TGTalker, the LLM directly outputs the destination node.

**Temporal Link Explanation with LLMs.** LLMs have been shown as powerful tools for explanability for tasks such as medical diagnosis analysis (Zhao et al., 2024), financial decision transparency (Limonad et al., 2024) and educational feedback (Seo et al., 2025). With LLMs directly interfacing with discrete language tokens, it is possible to generate temporal link explanations in natural language. The temporal link explanation task can also be formulated as a question-and-answer task for LLMs as follows:

$$\mathbf{R} = f^*(\mathbf{A}, g(\mathcal{G}_t), q(\mathbf{Q}_{s,?,t'})) \tag{2}$$

where $\mathbf{A} = f(g(\mathcal{G}_t), q(\mathbf{Q}_{s,?,t'}))$ is the answer to query $q(\mathbf{Q}_{s,?,t'})$ and $\mathbf{R}$ is the explanation or the reasoning for the answer. Note that the LLM $f^*$ used for the explanation $\mathbf{R}$ does not necessarily have to be the same LLM $f$ used for prediction. Explaining predictions on temporal graphs is inherently challenging due to the spatial and temporal dependencies between entities (Xia et al., 2022; Chen & Ying, 2023). Unlike standard TGNNs where only a prediction probability is produced, LLMs provide an alternative approach for contextualizing structural interactions via natural language.

**Pre-trained LLMs.** The TGTalker framework can be applied to any pre-trained LLM model. In this work, we consider 5 families of LLM models include `Qwen3` (Yang et al., 2024b), `Qwen2.5` (Yang et al., 2024b;a), `Mistral` (Chaplot et al., 2023), `Llama3` (Dubey et al., 2024) and `GPT-4.1-mini` (Achiam et al., 2023). Unlike temporal graph neural networks (TGNNs) that are trained from scratch for each dataset, pre-trained LLMs have high adaptability. With TGTalker, LLMs can be directly adapted to new TG datasets without any training or fine-tuning steps, significantly reducing the overhead for manual model selection and hyperparameter tuning. Lastly, TGTalker can inherently benefit from future improvements in language models' reasoning capacities.

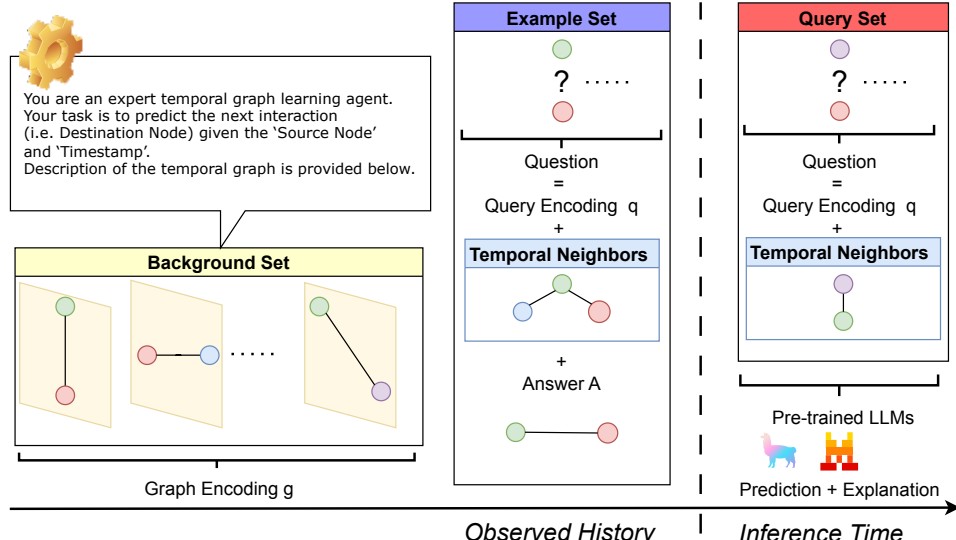

Figure 2: TGTalker framework has four core components: a background set, an example set, a query set and temporal neighbor sampling.

## 4.2 TEMPORAL GRAPH TALKER (TGTALKER)

Here, we discuss how TGTalker encodes a temporal graph for LLMs. TGTalker provides effective temporal graph encoding function $g$ and query encoding function $q$ for LLMs. Note that LLMs each have a fixed maximum input size $l_{\max}$, which limits the amount of temporal graph structural information that a LLM can receive. To tackle this issue, we leverage the strong recency bias in temporal graph (Cornell et al., 2025; Souza et al., 2022; Huang et al., 2023b; Poursafaei et al., 2022) to subsample the edges that are closest to the time of interest.

Figure 2 shows the overview of TGTalker which contains four core components: a *background set*, a *example set*, a *query set*, and *temporal neighbor sampling*. The background set contains recent temporal graph structures and acts as the context for the LLM. The example set shows matching question and answer pairs that instructs the LLM to follow the text correctly. The query set contains the current batch of edges for prediction and then converts them to LLM queries via the query encoding function $q$. Importantly, temporal neighbor sampling identifies the most recent neighbors for each source node in the query and retrieves them to provide the LLM with additional context. We provide an example prompt in Appendix C.

**Temporal Neighbor Sampling.** Both TGNNs (Xu et al., 2020; Rossi et al., 2020) and temporal graph transformers (Yu et al., 2023) utilize temporal neighbor sampling as a key component in architecture design. In TGTalker, we also augment LLMs with temporal neighborhood information, following Definition 2. For a given node of interest $u$, we implement a one-hop neighbor sampler which tracks its neighborhood information $\mathcal{N}_u^{t,1}$ and includes the most recent $m$ neighbors in relation to the current timestamp $t$. In practice, we observe that the inclusion of temporal neighbor information significantly boosts LLM performance as it directly retrieves the most relevant information for LLM.

**Background Set.** The background set contains $b$ most recent edges in $\mathcal{G}_t$ where $t$ is the timestamp of prediction and acts as the temporal graph encoding function $g(\mathcal{G}_t)$. The LLM is instructed to be a temporal graph learning agent, and then each node is encoded with an integer encoding, directly stated with its node ID. Edges are represented with parentheses as (src node ID, dst node ID, timestamp). The timestamps are represented with integer encoding as well. Note that on continuous time dynamic graphs, novel nodes are slowly introduced over time via their first edge interaction. Thus, assigning node IDs to nodes based on their chronological sequence of appearance provides a principled way to encode nodes with integer encoding. It is possible that multiple nodes are introduced at the same timestamp; in this case, the node IDs are assigned based on the order of edges from the original dataset. For bipartite graphs, it is common to assign non-overlapping node IDs for nodes in different partitions. We follow this convention and nodes within each partition are assigned an ID based on chronological order.

**Query Set and Example Set.** Following the convention in (Huang et al., 2023b; Rossi et al., 2020), TGTalker predicts test edges in a fixed size batch. Each test edge for evaluation is then converted to text via the query encoding function $q(\mathbf{Q}_{s,?,t'})$ where $s$ and $t$ are the source node and timestamp of interest, respectively. Similar to the background set, we encode the edges with parentheses and state nodes with their integer node ID. To provide additional context on the source node as input to LLM, we utilize *temporal neighbor sampling* here to also include $m$ most recent neighbors of the source node in the prompt. This retrieves the most relevant information of the source node from past history. The example set provides in-context learning examples for LLMs to follow. Each example is a question and answer pair i.e. $(\mathbf{A}, \mathbf{Q})$, instructing the model to follow the same question and answer patterns. The examples are selected as the most recent $k$ observed edges before the prediction time, also incorporating the recency bias. In this way, the example set also reflect any recent changes in the graph distribution. For our experiments, we consider the example set size of 5, thus conducting 5-shot prompting on LLM.

## 5 Experiments

**Datasets.** Here, we evaluate TGTalker across five continuous-time dynamic graph datasets. Dataset details are described in Appendix E. The number of nodes in our experiments are up to 500x larger than the synthetic graphs used in prior work (Zhang et al., 2024b). The surprise index is defined as $surprise = \frac{|E_{\text{test}} \backslash E_{\text{train}}|}{E_{\text{test}}}$ (Poursafaei et al., 2022) which measures the proportion of unseen edges in the test set when compared to the training set. The `tgbl-wiki` dataset is from the Temporal Graph Benchmark (TGB) (Huang et al., 2023b) while the rest of datasets are from (Poursafaei et al., 2022).

**Evaluation.** We follow the evaluation protocol in TGB where the link prediction task is treated as a ranking problem and the goal is to rank the true positive edge higher than all negative samples. For `tgbl-wiki`, we use the TGB negative samples, while for other datasets, we follow TGB procedure (Huang et al., 2023b) to generate negative samples for evaluation, including 50% historical negatives and 50% random negatives. For TGTalker models, as seen in Section 4.1, LLMs directly generate the destination node ID. To form the output probability vector for MRR ranking evaluation, we use an indicator vector with the predicted destination labelled with 1.0 probability while the rest is set to 0. In this way, TGTalker LLMs can be fairly compared with existing TGNN approaches.

**Variants and Baselines.** We test five families of LLMs with varying sizes as pre-trained LLMs in TGTalker, including the following open-source LLM models: `Qwen3-1.7B`, `Qwen3-8B`, `Qwen2.5-7B-Instruct` (Yang et al., 2024b;a), `Mistral-7B-Instruct-v0.3` (Chaplot et al., 2023) and `Llama3-8B-instruct` (Dubey et al., 2024). We also include the closed-source model: `GPT-4.1-mini` (Achiam et al., 2023). All of the LLMs are instruct models thus better fit for in-context learning. We drop the instruct suffix later for brevity. By default, we use background set size $b = 300$, batch size 200, example set size 5, and two 1-hop temporal neighbors for each source node during prediction. An ablation study for TGTalker is found in Appendix G, we observe that the inclusion of temporal neighbors has the strongest impact on model performance. We also report the inference time for LLM models in Appendix H, we observe that on all datasets, the inference time of LLMs is less than a day. For temporal graph methods, we compare TGTalker with SOTA event-based methods including TNCN (Zhang et al., 2024a), TGN (Rossi et al., 2020), GraphMixer (Cong et al., 2022) and two variants of EdgeBank (Poursafaei et al., 2022). Following the implementation in UTG (Huang et al.), we also include three SOTA event-based methods, HTGN (Yang et al., 2021), GCLSTM (Chen et al., 2022) and GCN (Kipf & Welling, 2016).

### 5.1 Temporal Link Prediction Results

In this section, we evaluate the efficacy of TGTalker for temporal link prediction against established TGL methods. Table 1 presents the test Mean Reciprocal Rank (MRR) for all models across five temporal networks. As TGTalker leverages pre-trained LLMs without task-specific fine-tuning, we report its results from a single inference run. For conventional TGL methods, which are trained from scratch for each dataset, we report the mean and standard deviation across five random seeds. The deterministic EdgeBank baseline is reported by a single run. In Table 1, TGTalker consistently achieves top-two performance on three out of five datasets and secures a top-three rank on the Enron and LastFM datasets when compared with SOTA TGL methods. This highlights the significant

Table 1: Test MRR comparison for temporal link prediction, results reported from 5 runs for TGNNs and a single run for pre-trained LLMs. All LLMs are instruct models. Top results are highlighted by **first**, second, *third*.

| | Method | tgbl-wiki | Reddit | LastFM | UCI | Enron |
|---|---|---|---|---|---|---|
| **TGTalker (ours)** | Qwen3-1.7B | 0.607 | 0.606 | 0.066 | 0.166 | 0.166 |
| | Qwen3-8B | *0.651* | 0.626 | 0.069 | 0.189 | 0.192 |
| | Qwen2.5-7B | 0.648 | 0.617 | *0.079* | *0.220* | 0.200 |
| | Mistral-7B-v0.3 | 0.604 | 0.612 | 0.067 | 0.212 | 0.184 |
| | Llama3-8B | 0.604 | 0.613 | 0.069 | 0.213 | 0.193 |
| | GPT-4.1-mini | 0.679 | *0.623* | 0.078 | 0.232 | *0.207* |
| **Event** | TGN | 0.396 ± 0.060 | 0.499 ± 0.011 | 0.053 ± 0.015 | 0.051 ± 0.011 | 0.130 ± 0.066 |
| | TNCN | **0.711** ± 0.007 | **0.696** ± 0.020 | **0.156** ± 0.002 | **0.245** ± 0.040 | **0.379** ± 0.081 |
| | GraphMixer | 0.118 ± 0.002 | 0.136 ± 0.078 | 0.087 ± 0.005 | 0.034 ± 0.062 | 0.014 ± 0.002 |
| | EdgeBank$_\infty$ | 0.495 | 0.485 | 0.020 | 0.079 | 0.101 |
| | EdgeBank$_{tw}$ | 0.600 | 0.589 | 0.026 | 0.222 | 0.141 |
| **Snapshot** | HTGN (UTG) | 0.464 ± 0.005 | 0.533 ± 0.007 | 0.027 ± 0.007 | 0.038 ± 0.005 | 0.107 ± 0.009 |
| | GCLSTM (UTG) | 0.374 ± 0.010 | 0.467 ± 0.004 | 0.019 ± 0.001 | 0.013 ± 0.001 | 0.157 ± 0.019 |
| | GCN (UTG) | 0.336 ± 0.009 | 0.242 ± 0.005 | 0.024 ± 0.001 | 0.022 ± 0.001 | 0.232 ± 0.005 |

potential of leveraging LLMs for complex temporal graph prediction tasks. In addition, TGTalker requires no fine-tuning or training when compared to existing methods.

Another key observation is that increased model size within the same LLM family boosts performance. For instance, Qwen3-8B generally outperforms Qwen3-1.7B across multiple datasets. However, this trend does not universally apply when comparing across different LLM families; for example, GPT-4.1-mini, despite its larger parameter count compared to some open-source models, exhibits notably lower performance than Qwen3-1.7B on several datasets. Intriguingly, the closed-source GPT-4.1-mini often underperforms its open-source counterparts of comparable or even smaller sizes, particularly on datasets like tgbl-wiki and Reddit. Among the LLM models, Qwen2.5-7B achieves the best overall performance. TGTalker processes only a recent subset of the temporal graph's history due to maximum token length limitations on LLMs. In contrast, conventional TGL methods are typically exposed to the entire graph history up to the prediction time. The competitive performance of TGTalker suggests that LLMs utilize the most salient recent structural and temporal information from the background set, example set and temporal neighbors in the TGTalker prompts.

## 5.2 TEMPORAL LINK EXPLANATION RESULTS

**Designing Explanation Categories.** To systematically analyze the diverse explanations, we first establish a set of reasoning categories. We first use Qwen3-8B and GPT-4.1-mini to generate example link explanations, and we also ask them to design appropriate categories for explanations. Then we retrieve the categories and perform human annotation to ensure a set of ten reasonable and comprehensive categories. These categories, along with their detailed descriptions, are shown in Table 2. We also provide concise example explanations from GPT-4.1-mini for each category. [1] Due to space constraints, full examples are provided in Appendix I. Our analysis reveals that LLMs can construct interesting explanations. The category of 'Most Recent Interaction' is similar to the well-known EdgeBank algorithm (Poursafaei et al., 2022) where the model simply uses recent past neighbors of a node for prediction. The 'Default or Most Common Node' category is analogous to the TG algorithm named PopTrack (Daniluk & Dabrowski) which captures recently globally popular destination nodes. In the provided reasoning, GPT-4.1-mini also defaults to the most frequent destination node. Another interesting set of explanations, such as 'Pattern Continuation' and 'sequence logic' involves pattern extrapolation and analyzing alternating interactions from a given source node to various destination nodes. These explanations show that LLMs can naturally discover algorithms on a temporal graph and provide insight into link pattern discovery on temporal networks.

**Quantitative Analysis of Explanation Categories.** To quantify the prevalence of these reasoning categories, we prompted the same LLM that generated the prediction to classify its explanation into

---

[1] We show GPT-4.1-mini examples because its explanations cover all the explanation categories.

Table 2: Categories of Temporal Link Explanations of LLMs with example explanations.

| Category & Description | `GPT-4.1-mini` Example |
|---|---|
| **Most Recent Interaction**: the model predicts the destination node as the one with which the source node had its most recent interaction before (or closest to) the given timestamp. | Considering the latest timestamp 2249408 is close in temporal sequence to the previous interactions, the most likely destination node remains 9180. |
| **Repeated Interaction Pattern**: if a source node has repeatedly interacted with the same destination node at multiple timestamps, the model predicts that this pattern will continue. | The source node 415 has two past interactions, both with the destination node 9180 at timestamps 2129703 and 2218092. This indicates a strong and consistent pattern of interaction with the destination node 9180. |
| **Most Frequent Past Destination**: when multiple past interactions exist, the explanation chooses the destination node that appears most frequently in the interaction history. | Identify the most frequent Destination Nodes that 4946 has interacted with close to the query timestamp 2237653. Based on past interaction frequency and recency, predict the next likely Destination Node for Source Node 4946 at the given timestamp 2237653. |
| **Pattern Continuation**: The model infers the next likely destination by extrapolating from observed interaction patterns, even when the exact match isn't present. | There is no direct information about interactions involving Source Node 969 in the given temporal graph dataset. However, analyzing the dataset, node 1546 connected to node 2093 repeatedly suggests that node 2093 is a likely target for nodes connected with 1546, an adjacent node to 969 in the provided history. Given no other direct interactions for 969, the best estimate based on similar patterns is that 969 will interact with destination node 2093 at timestamp 7554157, consistent with historical interactions of the related node 1546. |
| **Sequence or Alternation Logic**: the model uses the order of interactions (e.g., alternating between nodes) to predict the next likely destination. | The most recent interactions for node 431 are at timestamps 9168862 (to 3662) and 9168862 (approx) to 2461. Both are recent and close in time, and both destination nodes appear recurrently. Given that node 431 alternates between these two destination nodes, and there is no recent interaction indicating a new destination, the most likely next destination node at timestamp 9259228 is one of these two. Between 2461 and 3662: Given that there are slightly more occurrences of interactions to 3662, including one of the most recent timestamps (9168862), it is more likely the next interaction will also be to node 3662. |
| **Default or Most Common Node**: in the absence of a clear match, the explanation may default to the most common or logical node, or state that any node could be chosen. | Since no direct interactions for Source Node 1741 are present, and no pattern can be derived, we consider the most frequent Destination Node overall or a default prediction. The most frequent Destination Node connected to various sources in the temporal graph is 3692. |
| **Lack of Data**: when no clear pattern or sufficient data is available, the model defaults to a plausible guess, sometimes stating the lack of information. | No interactions are shown in the provided temporal graph for node 1624, so we have no historical interactions for node 1624 to infer from. |
| **New Node**: the model infers that the next interaction might be with a new node that hasn't appeared in the source node's history, especially if all previous interactions are exhausted. | The 'Source Node' 997 is not present in the provided TEMPORAL GRAPH. Thus, we must infer based on the context: The 'Source Node' 997 likely behaves similarly to node 1546, as their historical interactions from the user mention are similar. |
| **Ambiguous Candidates**: the explanation discusses more than one plausible destination (e.g., similar timestamps), and may use additional heuristics to select among them. | Analyze the frequency and recency of destination nodes for Source Node 1543. The latest interactions for Source Node 1543 before timestamp 10383318 are (1543, 3115, 10286093) and (1543, 2539, 10286406). Among these, 3115 and 2539 are recent and likely candidates. |
| **Others**: use this only if none of the above apply. Include a proposed new category name and brief justification in the required format. | Examine the provided past interactions for Source Node 2115. Both interactions are from node 6545 to 9180 at different timestamps. There's no direct interaction from 2115 in the given data, but observing the pattern for 6545 might help. Since there's no direct data on 2115 in the provided temporal graph, we look at the past interactions' pattern for node 6545 to infer the destination node for 2115 at the given timestamp. |

one of the ten established categories, including an 'Others' option for unclassifiable explanations. The same LLM is used for prediction and explanation classification because only the prediction LLM knows the reasoning for its prediction, while other LLMs might not give the same reasoning. We study the first 5,000 test predictions for `LLama3-8B` and `GPT-4.1-mini` on the `tgbl-wiki` and UCI datasets. Figure 3 top row illustrates the distribution of these explanation categories. A key finding is the adaptability of LLM reasoning to dataset characteristics: the composition of explanations varies significantly across datasets. For example, when comparing `GPT-4.1-mini`'s explanations on `tgbl-wiki` versus UCI, there is a marked increase in the 'Lack of Data' category for UCI. This aligns with UCI's known characteristic of introducing more novel nodes and interactions over time, contrasting with `tgbl-wiki`'s higher prevalence of repeating edges. This demonstrates that LLMs can tailor their reasoning strategies to the underlying data distribution.

Further comparison between LLMs in Figure 3 reveals distinct reasoning profiles. For instance, `LLama3-8B` explanations feature a higher proportion of the 'Pattern Continuation' category compared to `GPT-4.1-mini`, which, in turn, exhibits a greater percentage of explanations in the 'Repeated Interaction Pattern' category. These differences offer insights into the varying intrinsic reasoning capabilities and biases of different LLM architectures when applied to temporal graphs. Interestingly, this categorization also surfaces model-specific behaviors and potential limitations. For example, `Llama3-8B` puts none of the explanations in the 'Sequence or Alternation Logic' and 'Ambiguous Candidates' categories, and in some cases, explanations classified under 'Most Common Node' appeared to be hallucinations. Conversely, `GPT-4.1-mini` occasionally produced explanations categorized as 'Others' that suggested novel reasoning patterns, such as 'Analogy-Based Inference from Similar Node,' highlighting the potential for LLMs to uncover reasoning patterns.

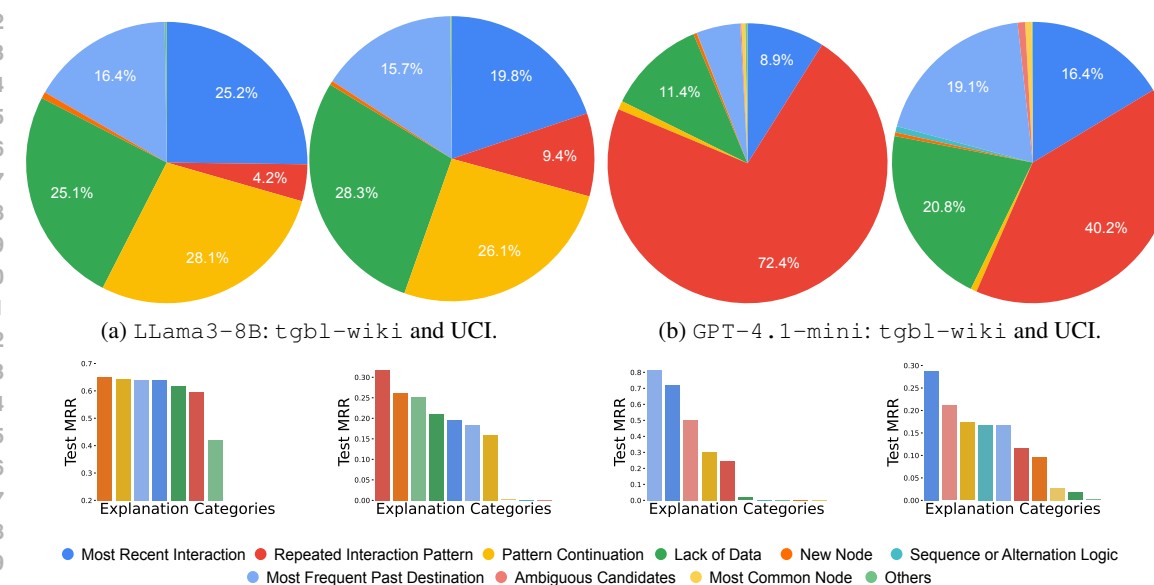

Figure 3: Explanation category composition (top row) and test MRR per category (bottom row) plots for `Llama3-8B` and `GPT-4.1-mini` on `tgbl-wiki` and UCI datasets. Note that empty explanation categories are not shown in bar plots.

Table 3: Human Annotation Experiment Results.

| Metric | `tgbl-wiki` | **UCI** |
|---|---|---|
| Avg. Correctness Rating (1–3) | 2.854 | 2.835 |
| Human–LLM Category: Raw Agreement | 70.83% | 72.00% |
| Intra-Annotator Agreement (Correctness Rating) | 75.00% | 74.00% |
| Intra-Annotator Agreement (Category Attribution) | 45.83% | 62.00% |
| Avg. Count of hallucination behaviour instances | 3% | 5.5% |

**Correlating Explanation Categories with Predictive Performance.** To assess whether the explanation was hallucinated or aligned with the predictive outcomes, we analyzed the test MRR performance within each category in Figure 3 bottom row. For `GPT-4.1-mini`, the results largely align with expectations: the 'Lack of Data' category consistently shows low MRR, validating that the LLM recognizes instances with insufficient evidence. Conversely, categories such as 'Most Recent Interaction' and 'Most Frequent Past Destination' exhibit high MRR, reinforcing the importance of recency and frequency heuristics in these temporal graphs. In comparison, `Llama3-8B` shows the strongest performance in 'Repeated Interaction Pattern' and 'Ambiguous Candidates' as well as surprisingly high performance in 'Lack of Data' categories, suggesting that `Llama3-8B` is less capable of correlating its explanation with likely predictive outcomes.

**Human Annotation Experiment.** To empirically validate the correctness of generated explanations, we conducted a human annotation experiment. Four independent annotators were asked to assess the correctness of the explanations and categorize them, as well as flag any instances of hallucinations. Table 3 details the results of this experiment. Notably we find LLM-generated explanations are rated very correct with a low hallucination rate, and considerable agreement between human-attributed categories and those generated by LLMs. We overall find high intra-annotator agreement to support these findings, though it is lower for category attribution, which we attribute to the high number of categories which induce higher variance.

## 6 CONCLUSION

In this work, we introduced TGTalker, a novel framework that leverages LLMs for prediction on real-world temporal graphs. TGTalker utilizes the recency bias inherent in temporal graphs and temporal neighbor sampling to extract the most relevant temporal graph structures to textual representations for LLM reasoning. TGTalker achieves competitive performance with state-of-the-art TGNNs across five

temporal networks without any fine-tuning. Beyond prediction accuracy, our framework introduces novel temporal link explanations via LLMs' natural language generation capabilities, providing human-readable explanations for predicted temporal links. The framework's ability to identify various reasoning patterns - from temporal relations to repeated patterns and sequence identification - opens new possibilities for explainability in temporal graph learning.

## REPRODUCIBILITY STATEMENT

We provide a complete, anonymized codebase to enable full reproducibility at `https://anonymous.4open.science/r/TGTalker-3FDB/README.md`. We also include the code-base in the supplementary materials. Access links to all open-source LLMs are included in Appendix F. Dataset details and download links are provided in Appendix E. Compute resources needed to reproduce the experiments are detailed in Appendix D.

## ETHICS STATEMENT

In this work, we propose the TGTalker framework to explore the use of LLM for temporal link prediction on real-world temporal graphs. It is well-known that LLMs can hallucinate and provide unreliable explanations (Ji et al., 2023; McKenna et al., 2023). As TGTalker framework utilizes pre-trained LLM, any hallucination or errors from the base LLM might also affect the output from TGTalker. Therefore, practitioners should be aware of the limitations of LLMs when adapting TGTalker. In addition, if the link explanation categories become widely-used in the field, it might limit novel categories to be discovered. We also expect that with novel temporal networks, more explanation categories will be discovered as LLMs adapts to more diverse link patterns.

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

## A    LLM USAGE

We acknowledge the use of LLMs to assist in polishing the writing of this paper. All content, ideas, figure and experimental results are our own. The LLM helped improve clarity and grammar.

## B    LIMITATIONS

First, the process of converting temporal graphs into textual representations is inherently constrained by the context window size of the underlying LLM. Given that the context window of LLMs are often limited to around 32k (small models) or 128k (larger models) for common open-source models such as Qwen (Yang et al., 2024b;a), it is not feasible to use entire temporal graphs with millions of edges as input. TGTalker is designed with this limitation in mind where we provide recent edges to the LLM in the input prompt. However, this strategy may overlook long-range dependencies(Dwivedi et al., 2022), which may lead to lower performance on datasets where relevant links span longer durations. Second, TGTalker is a framework for adapting LLMs for predictions on temporal graphs. Thus, the speed and efficiency of TGTalker are inherently tied to the characteristics of the base LLM model it employs. In addition, the choice of the base model affects link prediction performance as well as the explanations generated. Therefore, more LLM base models can be explored for evaluation as well.

## C    EXAMPLE TGTALKER PROMPT

System: You are an expert temporal graph learning agent. Your task is to predict the next interaction (i.e. Destination Node) given the Source Node and Timestamp.

Description of the temporal graph is provided below, where each line is a tuple of (Source Node, Destination Node, Timestamp).

TEMPORAL GRAPH:
(1478, 8773, 2270982)
(1555, 8789, 2270993)
...
# omitting edges due to space

# add example question answer pairs here

user: Source Node 179 has the following past interactions:
(179, 8994, 2129703)
(179, 8994, 2218092)

Please predict the most likely Destination Node for Source Node 179

at Timestamp 2272475

Figure 4: Example Prompt for TGTalker.

Table 4: Dataset statistics.

| Dataset | # Nodes | # Edges | # Unique Edges | # Unique Steps | Surprise | Duration |
|---------|---------|---------|----------------|----------------|----------|----------|
| `tgbl-wiki` | 9,227 | 157,474 | 18,257 | 152,757 | 0.108 | 1 month |
| Reddit | 10,984 | 672,447 | 78,516 | 669,065 | 0.069 | 1 month |
| LastFM | 1,980 | 1,293,103 | 154,993 | 1,283,614 | 0.35 | 1 month |
| UCI | 1,899 | 26,628 | 20,296 | 58,911 | 0.535 | 196 days |
| Enron | 184 | 10,472 | 3,125 | 22,632 | 0.253 | 3 years |

Figure 4 shows an example prompt for TGTalker.

## D  COMPUTE RESOURCES

For our experiments, we used the following compute resources. The TGTalker (LLM experiments) were conducted on a single NVIDIA A100-SXM4 GPU (80GB memory) paired with 4 AMD Milan 7413 CPU nodes (2.65 GHz, 128MB L3 cache), each equipped with 128GB RAM. For CTDGs and DTDGs experiments, we used a Quadro RTX 8000 GPU paired with 4 CPU nodes, each with 64GB RAM. Each experiment had a five-day time limit and was repeated five times, with results reported as averages and standard deviations across runs. For GPT-4o-mini and GPT-4.1-mini experiments, we used OpenAI's batch processing API `https://platform.openai.com/docs/guides/batch`. Aside from methods based on PyTorch Geometric, several baseline models, tested using their original code or the DyGLib repository, failed with out-of-memory or timeout errors on larger datasets, even under generous configurations.

## E  DATASET DETAILS

In this work, we conduct experiments on `tgbl-wiki`, Reddit, LastFM, UCI and Enron datasets. These datasets span a variety of real-world domains, providing a broad testbed for evaluating temporal graph models. Table 4 shows the statistics for each dataset. Our experiments cover a wide range of datasets with up to 1.2 million edges and timestamps.

- `tgbl-wiki` (Huang et al., 2023b) is a bipartite interaction network that captures temporal editing activity on Wikipedia over one month. The nodes represent Wikipedia pages and their editors, and the edges indicate timestamped edits. Each edge is a 172-dimensional LIWC feature vector derived from the edited text.

- **Reddit** (Poursafaei et al., 2022) models user-subreddit posting behavior over one month. Nodes are users and subreddits, and edges represent posting requests made by users to subreddits, each associated with a timestamp. Each edge is also a 172-dimensional LIWC feature vector based on post contents.

- **LastFM** (Poursafaei et al., 2022) is a bipartite user–item interaction graph where nodes represent users and songs. Edges indicate that a user listened to a particular song at a given time. The dataset includes 1000 users and the 1000 most-listened songs over a one-month period. It contains no node or edge attributes.

- **UCI** (Poursafaei et al., 2022) is an anonymized online social network from the University of California, Irvine. Nodes represent students, and edges represent timestamped private messages exchanged within an online student community. The dataset does not contain node or edge attributes.

- **Enron** (Poursafaei et al., 2022) is a temporal communication network that is based on email correspondence over a period of three years. Nodes represent employees of the ENRON energy company, while edges correspond to timestamped emails. The dataset does not include node or edge features.

`Tgbl-wiki` is obtained from the Temporal Graph Benchmark (Huang et al., 2023b), where the dataset can be downloaded along with the package, see TGB website for details. Reddit, LastFM, UCI and Enron are obtained from (Poursafaei et al., 2022) and can be downloaded at `https://zenodo.org/records/7213796#.Y8QicOzMJB2`.

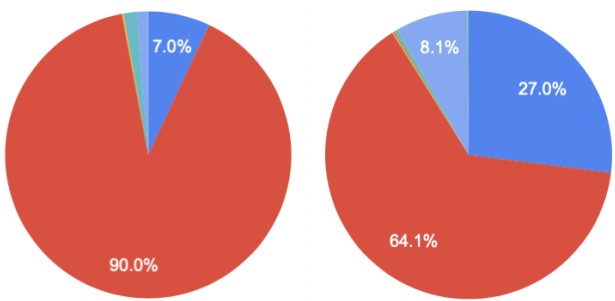

Figure 5: Explanation category composition plots for `Qwen2.5-7B` on tgbl-wiki and UCI datasets.

## F   CODE AND MODEL ACCESS

Code for reproducing TGTalker is available at `https://anonymous.4open.science/r/`
`TGTalker-3FDB/README.md`. We use various pre-trained LLMs in TGTalker. For reproducibility, the Hugging Face link to all the open-source LLMs are available below:

- `Qwen3-1.7B`: `https://huggingface.co/Qwen/Qwen3-1.7B`

- `Qwen3-8B`: `https://huggingface.co/Qwen/Qwen3-8B`

- `Qwen2.5-7B-Instruct`: `https://huggingface.co/Qwen/Qwen2.5-7B-Instruct`

- `Mistral-7B-Instruct-v0.3`: `https://huggingface.co/mistralai/Mistral-7B-Instruct-v0.3`

- `Llama3-8B-instruct`: `https://huggingface.co/meta-llama/Meta-Llama-3-8B-Instruct`

## G   ABLATION STUDY

Here we conduct ablation studies on the components of TGTalker. Table 5 shows the link prediction performance of the `Qwen3.8b` model with respect to *in context learning*, the number of *temporal neighbors*, and the size of the *background set* – as well as the performance of TGTalker with none of these components. Notably, the exclusion of temporal neighbours has a significant reduction in model performance – and the exclusion of all three components drastically nullifies the model's performance. This emphasizes the importance of extracting relevant structural information specific to the node of interest (i.e. the recent neighbours of the source node).

Table 5: Ablation Study Results for TGTalker with `Qwen3.8b` as base model where test MRR is reported.

| Configuration | tgbl-wiki | Reddit |
|---|---|---|
| TGTalker (All components) | 0.649 | 0.613 |
| – w/o In Context Learning | 0.645 | 0.612 |
| – w/o Temporal Neighbours | 0.322 | 0.122 |
| – w/o Background Set | 0.648 | 0.618 |
| – with no components | 0.008 | 0.002 |

In comparison, removing either the In-context learning and background set seems to have minor impact on the model performance. Their importance is however put forth by the drop in performance with no components. Table 6 offers more detailed insights into the effect of temporal neighbours. While the absence of such neighbours drastically plummets TGTalker's performance, the increase in number of neighbours also has a strong positive effect.

Table 6: Ablation Study Results for TGTalker with `Qwen3.8b` on the number of temporal neighbors

| Configuration | `tgbl-wiki` | Reddit |
|---|---|---|
| TGTalker (nbr = 2) | 0.649 | 0.613 |
| – Number of temporal neighbours = 0 | 0.322 | 0.122 |
| – Number of temporal neighbours = 1 | 0.646 | 0.520 |
| – Number of temporal neighbours = 5 | 0.652 | 0.635 |
| – Number of temporal neighbours = 10 | 0.656 | 0.643 |

## H  LLM INFERENCE TIME

Table 7 report the inference time of both `Qwen3-8B` and `Llama3-8B` across our benchmark datasets. We report directly the inference time on the test set for temporal link prediction with an NVIDIA Quadro RTX 8000 GPU (48GB). We observe that on all datasets, the inference time of LLMs is less than a day. In particular, most datasets like `tgbl-wiki`, UCI and Enron only have inference time of a few hours.

Table 7: Inference time cost of LLM models

| Dataset | Model | Inference Time (secs) | Approx. Hours |
|---|---|---|---|
| `tgbl-wiki` | `Qwen3-8B` | 10773 | 3 |
| `tgbl-wiki` | `Llama3-8B` | 10377 | 3 |
| Reddit | `Qwen3-8B` | 50297 | 14 |
| Reddit | `Llama3-8B` | 45137 | 13 |
| LastFM | `Qwen3-8B` | 86025 | 23 |
| LastFM | `Llama3-8B` | 84341 | 23 |
| UCI | `Qwen3-8B` | 4046 | 1 |
| UCI | `Llama3-8B` | 3839 | 1 |
| Enron | `Qwen3-8B` | 7755 | 2 |
| Enron | `Llama3-8B` | 7304 | 2 |

# I    MODEL EXPLANATIONS

Here we provide further details on the temporal link explanations generated by LLMs. Table 2 presents all ten categories of explanations along with their corresponding descriptions. The 'Other' category is left open for the LLM to propose novel categories if desired. Figure 5 illustrates the distribution of explanations generated by `Qwen2.5-7B`, highlighting the model's preference for the *Repeated Interaction Pattern* category. Example explanations from `Llama3-8B-instruct`, `GPT-4.1-mini`, and `Qwen2.5-7B` are shown in Table 8, Table 10, and Table 9, respectively. Notably, `Llama3-8B-instruct` omits explanations in certain categories such as 'Ambiguous Candidates' and exhibits hallucinations in the 'Default or Most Common Node' category. Similarly, `Qwen2.5-7B` often misclassifies its 'Default' hallucinations, which more accurately reflect 'Lack of Data' descriptions.

Table 8: Full Examples of 'GPT-4.1-mini' explanations for each category.

| Category Description | `GPT-4.1-mini` Example |
|---|---|
| **Most Recent Interaction**: The model predicts the destination node as the one with which the source node had its most recent interaction before (or closest to) the given timestamp. | Given the past interactions of Source Node 5124, it has had interactions only with Destination Node 9180 at timestamps 2129703 and 2218092. There is no evidence of any other destination node for this source node. Considering the latest timestamp 2249408 is close in temporal sequence to the previous interactions, the most likely destination node remains 9180. |
| **Repeated Interaction Pattern**: if a source node has repeatedly interacted with the same destination node at multiple timestamps, the model predicts that this pattern will continue. | The source node 415 has two past interactions, both with the destination node 9180 at timestamps 2129703 and 2218092. This indicates a strong and consistent pattern of interaction with the destination node 9180.The latest past interaction timestamp 2218092 is earlier than the query timestamp 2238574, so the pattern is likely to continue. |
| **Most Frequent Past Destination**: When multiple past interactions exist, the explanation chooses the destination node that appears most frequently in the interaction history. | Analyze the historical interactions for Source Node 4946 in the provided temporal graph data to identify any patterns or repeated Destination Nodes. Identify the most frequent Destination Nodes that 4946 has interacted with close to the query timestamp 2237653. Based on past interaction frequency and recency, predict the next likely Destination Node for Source Node 4946 at the given timestamp 2237653. |
| **Pattern Continuation**: The model infers the next likely destination by extrapolating from observed interaction patterns, even when the exact match isn't present. | The interaction history of Source Node 969 shows edges from node 1546 to node 2093 at timestamps 5074383 and 5116440. There is no direct information about interactions involving Source Node 969 in the given temporal graph dataset. However, analyzing the dataset, node 1546 connected to node 2093 repeatedly suggests that node 2093 is a likely target for nodes connected with 1546, an adjacent node to 969 in the provided history. Given no other direct interactions for 969, the best estimate based on similar patterns is that 969 will interact with destination node 2093 at timestamp 7554157, consistent with historical interactions of the related node 1546. |
| **Sequence or Alternation Logic**: The model uses the order of interactions (e.g., alternating between nodes) to predict the next likely destination. | Given the provided temporal graph, we search for interactions involving Source Node 431. We observe multiple interactions from node 431 to destination nodes 2461 and 3662 at various times, e.g. (431, 2461, 9031465), (431, 3662, 9041425), (431, 3662, 9090133), (431, 2461, 9090545), (431, 3662, 9168862). This indicates that node 431 interacts mainly with nodes 2461 and 3662. Comparing the timestamp 9259228 to the latest interactions, the most recent interactions for node 431 are at timestamps 9168862 (to 3662) and 9168862 (approx) to 2461. Both are recent and close in time, and both destination nodes appear recurrently. Given that node 431 alternates between these two destination nodes, and there is no recent interaction indicating a new destination, the most likely next destination node at timestamp 9259228 is one of these two. Between 2461 and 3662: Given that there are slightly more occurrences of interactions to 3662, including one of the most recent timestamps (9168862), it is more likely the next interaction will also be to node 3662. |
| **Default or Most Common Node**: In the absence of a clear match, the explanation may default to the most common or logical node, or state that any node could be chosen. | Check the temporal graph for interactions involving Source Node 1741. None are found in the given data, so no direct pattern exists for this node. Examine the given past interactions of Source Node 1741: (1546, 2093, 5074383) and (1546, 2093, 5116440). These involve Source Node 1546 connecting to Destination Node 2093, but Source Node 1741 itself is not involved. Possibly a misunderstanding or misstatement of data. Since no direct interactions for Source Node 1741 are present, and no pattern can be derived, we consider the most frequent Destination Node overall or a default prediction. The most frequent Destination Node connected to various sources in the temporal graph is 3692. |
| **Lack of Data**: When no clear pattern or sufficient data is available, the model defaults to a plausible guess, sometimes stating the lack of information. | The given past interactions for Source Node 1624 are (1546, 2093, 5074383) and (1546, 2093, 5116440). However, these are for Source Node 1546, not 1624. No interactions are shown in the provided temporal graph for node 1624, so we have no historical interactions for node 1624 to infer from. |
| **New Node**: The model infers that the next interaction might be with a new node that hasn't appeared in the source node's history, especially if all previous interactions are exhausted. | The 'Source Node' 997 is not present in the provided TEMPORAL GRAPH. Instead, the user mentioned past interactions with (1546, 2093) at lower timestamps, but these do not relate to 'Source Node' 997 directly in the current temporal graph data. Thus, we must infer based on the context: The 'Source Node' 997 likely behaves similarly to node 1546, as their historical interactions from the user mention are similar. |
| **Ambiguous Candidates**: The explanation discusses more than one plausible destination (e.g., similar timestamps), and may use additional heuristics to select among them. | Check the temporal graph for interactions where Source Node is 1543. Analyze the frequency and recency of destination nodes for Source Node 1543. The latest interactions for Source Node 1543 before timestamp 10383318 are (1543, 3115, 10286093) and (1543, 2539, 10286406). Among these, 3115 and 2539 are recent and likely candidates. |
| **Others**: Use this only if none of the above apply. Include a proposed new category name and brief justification in the required format. | Examine the provided past interactions for Source Node 2115. Both interactions are from node 6545 to 9180 at different timestamps. There's no direct interaction from 2115 in the given data, but observing the pattern for 6545 might help. Check the temporal graph to see any interactions involving 2115. Since there's no direct data on 2115 in the provided temporal graph, we look at the past interactions' pattern for node 6545 to infer the destination node for 2115 at the given timestamp. |

Table 9: Examples of `Qwen2.5-7B` explanations for each category.

| Category Description | `Qwen2.5-7B` Example |
|---|---|
| **Most Recent Interaction**: The model predicts the destination node as the one with which the source node had its most recent interaction before (or closest to) the given timestamp. | I used the temporal graph provided to analyze the movement patterns between Node 775 and its recorded neighbors in the graph. Given that Node 775's movement has been predominately towards Node 8529, and aiming to predict next interaction's node, the Node 8529 is predicted as the most plausible neighbor based on past interactions. |
| **Repeated Interaction Pattern**: If a source node has repeatedly interacted with the same destination node at multiple timestamps, the model predicts that this pattern will continue. | I used the temporal graph provided to analyze the movement patterns between Node 7326, Node 400, and Node 2517. Based on previous interactions, I determined the two most frequent Destination Nodes for each Source Node. Therefore, my prediction for Node 7326 is 8491, and my prediction for Nodes 400 and 2517 is 8817 respectively. |
| **Most Frequent Past Destination**: When multiple past interactions exist, the explanation chooses the destination node that appears most frequently in the interaction history. | I predict that the most likely 'Destination Node' for 'Source Node' 2196 at 'Timestamp' 2220367 is '8471' because it has been connected to this node more frequently in the temporal graph observations. The adjacency relationships between these nodes is the key factor in my prediction. |
| **Pattern Continuation**: The model infers the next likely destination by extrapolating from observed interaction patterns, even when the exact match isn't present. | I predict that the most likely destination node for Source Node 1282 at Timestamp 7206287 is 3185 based on its previous interactions. It has visited two other nodes, 1285 and 2315, around 79 times in Temporal Graph. This visit distribution across these three nodes is discovered by revisiting the Temporal Graph which fits well with a triadic closure of nodes 1282, 1285 and 2315 and highlights potential strength of three nodes to communicate with each other, indicating that 1282 might be looking to create new connections. Therefore, considering the node co-occurrence, node strength and high triadic closure, I predict Render Tree to be the destination node of 1282 at Timestamp 7206287. |
| **Sequence or Alternation Logic**: The model uses the order of interactions (e.g., alternating between nodes) to predict the next likely destination. | I used the temporal graph provided to analyze the movement patterns between Node 775 and its recorded neighbors in the graph. Given that Node 775's movement pattern changes between Node 9180 and Node 8529, I predict Node 8529 as the most likely next destination because it represents a different pattern from the one represented by Node 9180. |
| **Default or Most Common Node**: In the absence of a clear match, the explanation may default to the most common or logical node, or state that any node could be chosen. | The only explanation generated in this category is a hallucination confounded with the *Lack of Data* category. |
| **Lack of Data**: When no clear pattern or sufficient data is available, the model defaults to a plausible guess, sometimes stating the lack of information. | I used the temporal graph provided to analyze the movement patterns between Node 312 and its recorded neighbors in the graph. Given that Node 390 has existed in previous time-steps but is not seen connected to Node 2032, it can be inferred that Nodes 390 and 2032 are not directly connected to one another at this point. Therefore, predicting a connection between 312 and 2032 might not be accurate, and choosing them as a destination could be a random guess without further context. |
| **New Node**: The model infers that the next interaction might be with a new node that hasn't appeared in the source node's history, especially if all previous interactions are exhausted. | I predict that the most likely destination node for Source Node 1285 at Timestamp 7198235 is 3213 based on its previous interactions. It has visited two other nodes, 1282 and 3547, around 7091387. Since 3213 is not discovered in these past interactions and its occurrence in Temporal Graph is less than these two nodes, I believe it has not been visited yet. |
| **Ambiguous Candidates**: The explanation discusses more than one plausible destination (e.g., similar timestamps), and may use additional heuristics to select among them. | No explanations fall within this category (None). |
| **Others**: Use this only if none of the above apply. Include a proposed new category name and brief justification in the required format. | I predict that the most likely destination node for Source Node 447 at Timestamp 7138932 is 3567 due to my analysis of the temporal graph data, in which Source Node 447 has not interacted with any other Node in the provided data. Since Temporal Graph Learning is based on historical patterns, historical data discovered in the past, and interactions between nodes over time, I predict that no further interaction is likely to occur. Please note that this prediction is based purely on the pattern of absence in the provided data and Temporal Graph Learning best practices, and not on any other external factors. This prediction should be taken as a stochastic exercise and may not be entirely accurate always. |

Table 10: Examples of `Llama3-8B-instruct` explanations for each category.

| Category Description | `Llama3-8B-instruct` Example |
|---|---|
| **Most Recent Interaction**: The model predicts the destination node as the one with which the source node had its most recent interaction before (or closest to) the given timestamp. | The source node 131 has a past interaction with destination node 220, which is a relatively strong signal. Since there is no other strong signal in the data, I predict that the most likely destination node for source node 131 at timestamp 94823623 is node 276, which is the node that is closest to node 220 in the graph. |
| **Repeated Interaction Pattern**: If a source node has repeatedly interacted with the same destination node at multiple timestamps, the model predicts that this pattern will continue. | The temporal graph model learns patterns and relationships based on the past interactions. When we look at the past interactions for 'Source Node' 6, we see that it typically interacts with nodes 237 and 314. Considering the timestamp 94969339, it's possible that the pattern of interactions is maintained, and the most likely destination node for 'Source Node' 6 would be 'Node 314'. However, please note that this prediction is based on a limited set of past interactions and may not generalize to all possible future interactions. |
| **Most Frequent Past Destination**: When multiple past interactions exist, the explanation chooses the destination node that appears most frequently in the interaction history. | Based on the given temporal graph, which includes multiple interactions across different timestamps, I can see that 'Source Node' 147 has a history of interactions with several nodes, including 280, 293, 314, and 367. The predicted timestamp (95227379) is in the same time range as the majority of interactions. Therefore, I predict that the most likely 'Destination Node' for 'Source Node' 147 at 'Timestamp' 95227379 is node 367, given that this node has a high frequency of interactions with node 147 in the past. |
| **Pattern Continuation**: The model infers the next likely destination by extrapolating from observed interaction patterns, even when the exact match isn't present. | The temporal graph model learns to predict the likelihood of a future interaction based on past interactions. Since 'Source Node' 117 has only two past interactions, both with 'Destination Node' 64, which is not present in the general interaction list, it's difficult to determine a pattern or relationship. However, the closest pair to 'Source Node' 117 is 'Source Node' 125 with a time difference of 18, and its closest pair is a repetition of the same interaction. Therefore, I predicted 'Destination Node' 299 as it is similar in structure to the past interaction of 'Source Node' 125, even though it may not be the exact same node. |
| **Sequence or Alternation Logic**: The model uses the order of interactions (e.g., alternating between nodes) to predict the next likely destination. | No explanations fall within this category. (None) |
| **Default or Most Common Node**: In the absence of a clear match, the explanation may default to the most common or logical node, or state that any node could be chosen. | The only explanation in this category is a hallucination. |
| **Lack of Data**: When no clear pattern or sufficient data is available, the model defaults to a plausible guess, sometimes stating the lack of information. | Unfortunately, since there is no past interaction data available for 'Source Node' 2134, I cannot provide a reliable prediction. The graph learning model relies heavily on past interactions to make predictions, so without any relevant data, it's challenging to make an accurate prediction. If more data was available, I could train the model to predict the most likely 'Destination Node' for 'Source Node' 2134 at 'Timestamp' 2479023. |
| **New Node**: The model infers that the next interaction might be with a new node that hasn't appeared in the source node's history, especially if all previous interactions are exhausted. | The source node 1546 is not available in the previous network as we moved to source node 1548. The name sequences of the source node may differ but all other procedures remain same. |
| **Ambiguous Candidates**: The explanation discusses more than one plausible destination (e.g., similar timestamps), and may use additional heuristics to select among them. | No explanations fall within this category. (None) |
| **Others**: Use this only if none of the above apply. Include a proposed new category name and brief justification in the required format. | Unfortunately, since you provided a random input with fake interactions, I don't have enough real data to accurately predict the destination node. The interactions you provided have no existing pattern or correlation with any actual node or timestamp, so I would recommend trying with real data for a more accurate prediction. |

