# OpenReview forum: "Are Large Language Models Good Temporal Graph Learners?"
_ICLR.cc/2026/Conference — ICLR 2026 Conference Withdrawn Submission_

### Official Review · Reviewer_uJSF · 2025-10-30

**Soundness:** 3
**Presentation:** 2
**Contribution:** 2
**Rating:** 4
**Confidence:** 4

**Summary:**

The authors have introduced TGTalker, a framework that use LLMs to perform link prediction on temporal graphs and provide natural language explanations. The experiments are carried out on five dataset on temporal-graphs and while the performances reached are comparable to other methods, the fact that have been reached without fine-tuning or training the models is very important.

**Strengths:**

The main strengths of the work are the following:

- Strength 1: the paper is well-written, easy to follow and logically structured. Figures effectively illustrate the framework. As a very side note, numbers on Figure 3 may be a bit too small;

- Strength 2: the experiments span multiple LLMs and multiple datasets providing a solid empirical grounding;

- Strength 3: the work applies LLMs to TG and this is relevant to current trends in combining structured reasoning with language models.

**Weaknesses:**

The main weaknesses of this paper are the following ones:

Weaknesses 1: while the paper describes the context for TGTalker, there is not a proper discussion on why this specific design (e.g., background set, example set, …) works or how sensitive the results are to prompt formulation. There is also a limited discussion on computational constraints (e.g., token limits, context truncation strategies,...) and related impact

Weaknesses 2: concerning the selected baselines, it is not clear whether the same temporal splits and negative samplings are identical for all models. While it is clear for LLMs, are the baselines trained with the same level of data accessibility?

Weaknesses 3: it was interesting that the agreement scores in Table 3 are around the 70%. Any chance to have a deeper interpretation in the discussion of where the disagreements arise? any specific category? Also, some of the categories in the taxonomy presented in Table 2 show some overlapping (for example “Pattern Continuation” and “Repeated Interaction Patterns”).

Weaknesses 4: concerning the explanations and the human annotation, it would be important to understand on which models the alignments are computed. Are the values reported averages? If so, a standard deviation is missing. Otherwise, on which model’s explanation are the alignments computed? Is there an LLM that is better than others? which are the minimum and maximum alignments across the models? A proper evaluation and discussion is missing and, as the explanation is a key contribution of the paper, it would be important to produce a clearer table and more specific results and a deeper discussion.

**Questions:**

Some questions are the following ones:

See Table 3, where the disagreements arise? any specific category?

Concerning the explanations, on which model’s explanation are the alignments computed? Is there an LLM that is better than others? which are the minimum and maximum alignments across the models?

---

> ### Author Response · Authors · 2025-11-24
> **Author Response to Reviewer uJSF**
>
> We thank the reviewer for the time and consideration of our paper. We would like to address your comments below.
> ---
>
>
> **W1** Design justification
>
> > There is no proper discussion on why the TGTalker architecture works, or how sensitive the results are to prompt formulation.
>
> **Response.** We agree that the rationale behind the prompt components could be clarified. Our design choices are grounded in well-established temporal graph learning principles rather than ad-hoc formatting:
>
> - **Recency bias is a strong inductive bias.** TGTalker focus on incorporating recent edges related to the prediction as context for the LLM.
> - **Neighbor information is highly important.** The temporal neighbor sampling component in TGTalker provides important information for LLM reasoning to leverage most relevant past edges for the prediction.
>
>  While designing the TGTalker framework, we explored with the prompt formulation and component ordering and found that, though it has importance and different formulations can guide the LLM to have different behaviours, these changes are somewhat negligible and it seemed that the architecture itself was fundamentally responsible for the model’s performance.
>
>
> **W2** Baselines
>
> > It is not clear whether the same temporal splits and negative samplings are identical for all models. Are the baselines trained with the same level of data accessibility?
>
> **Response.** Thank you for this question. Yes, all the TGTalker LLMs and TGNN models follow the same temporal splits (train, val, test) as in the temporal graph benchmark. The negative samples are also generated in the same fashion as TGB thus ensuring the reproducibility of the evaluation. TGTalker requires no training and the models directly inference on the test data while TGNN models are trained on the training set. In terms of data accessibility, all models have access to information prior to the prediction timestamp such as for the purpose of temporal neighbor sampling.
>
>
> **W3** Agreement
>
> > Any chance to have a deeper interpretation of the agreement?
>
>
> **Response.** We agree the paper could’ve benefitted from this discussion. We interpreted these disagreements to mainly stem from the large category space. With a high number of categories, some of which can have overlapping characteristics, we expected some natural variance to emerge and thus believe the current agreement rates are reasonable with the multitude of choice that was presented to annotators.
>
>
> **W4** Human Annotation
>
> > Which model for human annotation?
>
> **Response.** We appreciate the reviewer’s thoroughness. The human annotation experiment, because quite costly in terms of time and human resources, was limited to one model, the GPT-4.1-mini. We chose this model as it is a recent one that is among the highest performing models we have evaluated. We agree with the fact that a detailed evaluation comparing alignment across models would be very interesting and will consider this as a direct next work. However at this point, we use GPT-4.1-mini as our key focus, and suspect findings from this model to at least somewhat generalize across other LLMs.
>
>
> **Q1**
>
> > Where do disagreements arise?
>
> **Response.** We appreciate the reviewer’s questions and re-iterate that we agree a comparison of agreement across models would be beneficial. In the meantime of finding a way to do this while dealing with the accompanying cost, we do not find any specific disagreement within a certain category, and instead attribute the overall disagreement to noisy variance inherent to the high number of categories. The 10 categories having overlapping features can indeed make selection ambiguous, which would also explain why the correctness ratings suffer from less disagreement.
>
>
> **Q2**
>
> > Which model is used to evaluate alignment? Which one is most aligned?
>
> **Response.**  The human annotation experiment, because quite costly in terms of time and human resources, was limited to one model, the GPT-4.1-mini. We chose this model as it is a recent one that is among the highest performing models we have evaluated. We agree with the fact that a detailed comparison across models would be very interesting and will consider this as a direct next step, and appreciate your suggestion.

---

### Official Review · Reviewer_tTdT · 2025-10-30

**Soundness:** 2
**Presentation:** 2
**Contribution:** 1
**Rating:** 2
**Confidence:** 5

**Summary:**

This paper extends the LLM4DyG (KDD'24) and utilizes LLMs to perform explainable temporal link prediction. The authors introduce a series of prompt engineering for the in-context learning of the LLM, including constructing the background (for showing the recent temporal graph interactions), examples (for few-shot learning), queries (for asking questions), and temporal neighbor sampling (for pointing out the recent neighborhoods for a given node). They also conduct experiments to validate the effectiveness of the proposed methods.

**Strengths:**

1. The authors introduce prompt engineering of the LLMs for the temporal link prediction tasks in a temporal graph, which is easy to follow.
2. The paper is well-written.

**Weaknesses:**

1. This paper seems to lack novelty. TGTalker employs a similar prompting paradigm to LLM4DyG, and no learning process has been introduced during the whole pipeline. Moreover, utilizing LLMs cannot be considered a novelty within the temporal graph learning community, as there already exist related works that integrate LLMs into temporal graphs [1][2].
2. The baselines used in this paper seem to be insufficient. The authors only compare against the recent model TNCN, while the other baselines are quite outdated. Compared to AP, MRR can be unstable when evaluating different models. Therefore, it is necessary to include more recent baselines, such as DyGFormer and NAT, to make the comparison more comprehensive and convincing.
3. The scalability of TGTalker may be a concern. The scales of datasets used in this paper are relatively small by the standards of the temporal graph learning community. For each given edge, the LLM should perform reasoning, and incur such computational costs are not affordable in the large datasets.
4. The design of the "background set" seems ineffective. As shown in Table 5, removing this module does not lead to any noticeable performance degradation; in fact, the performance even slightly improves.
5. MRR is a rank-based metric. How do you compute the ranking from the LLM’s generated responses?
6. Line 268 mentioned that TGTalker predicts test edges in fixed-size batches (under 200). However, Figure 4 only shows a single prompt example. How do you handle the prompt becoming excessively long, especially with such a batch size?

[1] LLM-driven Knowledge Distillation for Dynamic Text-attributed Graphs, AAAI 2025.
[2] Unifying Text Semantics and Graph Structures for Temporal Text-attributed Graphs with Large Language Models, NeurIPS 2025.

**Questions:**

See weaknesses.

---

> ### Author Response · Authors · 2025-11-24
> **Author Response to Reviewer tTdT**
>
> We thank the reviewer for the time and consideration of our paper. We would like to address your comments below.
> —
>
>
> **W1** novelty
>
> > This paper seems to lack novelty. TGTalker employs a similar prompting paradigm to LLM4DyG, and no learning process has been introduced during the whole pipeline. Moreover, utilizing LLMs cannot be considered a novelty within the temporal graph learning community, as there already exist related works that integrate LLMs into temporal graphs [1][2].
>
> **Response.** We respectfully disagree with the reviewer. The referenced prior work only focuses on text-attributed temporal graphs. It is well-known that LLMs has strong semantic information from its pre-training thus it is easy for LLM to work on text-attributed graphs as LLM can leverage the semantic information from text. In this work, the temporal graphs are anonymized and the LLM must reason over raw node IDs and edge timestamps focusing on their spatial and temporal reasoning capability. Therefore, TGTalker shows that LLMs can work well on general temporal graphs without relying on text attributes.
>
>
> **W2** Baselines
>
> > The baselines used in this paper seem to be insufficient, and MRR may be unstable.
>
> **Response.** We appreciate the reviewer’s thoroughness. However, MRR has become widely used for these tasks because, while AP may be stable, that stability may also be due to saturation; AP can very often achieve perfect accuracy and thus doesn’t effectively offer useful insights in understanding the state of the art. Would the reviewer kindly provide additional insight into why MRRs are unstable and the detailed advantages of using AP?
>
>
>
> **W3** Scalability
>
> > The scalability of TGTalker may be a concern.
>
> **Response.** We agree that larger datasets are important for real-world applications and appreciate the reviewer bringing attention to this. We note that TGTalker can work on larger graphs, because it operates on fixed-size sampled temporal neighborhoods, so prompt length is bounded even if the full graph has millions of nodes or edges. Thus, while we follow prior temporal graph benchmarks to select our datasets and have controlled comparison, the framework itself is, to the extent that we have evaluated, graph-size-agnostic.
>
>
> **W4** Background Set
>
> > The design of the background set is questionable.
>
> **Response.** While we may consider re-structuring the background set architecture, its effect as demonstrated in the ablation studies is insufficient to draw any conclusion, as it induces negligible changes in performance on both datasets, improving performance on one but degrading it on the other. To empirically motivate next steps, we will thus instead, conduct a more comprehensive ablation study that will cover all datasets as well as the 6 different underlying LLMs we explore using, and determine from those results, what the true trend is regarding the background set’s influence, and what design choices will then be appropriate.
>
>
> **W5** MRR as a rank-based metric
>
> > How do you compute a ranking from the LLM’s generated responses?
>
> **Response.** We warmly thank the reviewer for their question, for which we have clarified the corresponding sections in our paper ("Evaluation"). We clarify that LLM directly generates the destination node ID. To compare with existing TGNN approaches which outputs a probability vector ranking all destinations, we use an indicator vector to represent the LLM output, with the predicted destination labelled with 1.0 probability while the rest is set to 0. In this way, the MRR metric can be computed and then TGTalker LLMs can be compared fairly with existing TGNN approaches.
>
> **W6** Prompt length
>
> > How do you handle the prompt becoming excessively long, especially with such a batch size (200).
>
> **Response.** Although the script processes test edges in batches of up to 200 edges, it does not place 200 queries into a single prompt at any point; instead, the batching affects state updates. Thus the LLM sees exactly one query per call since it has one inference call per edge.

---

### Official Review · Reviewer_wyy6 · 2025-11-01

**Soundness:** 3
**Presentation:** 3
**Contribution:** 2
**Rating:** 2
**Confidence:** 4

**Summary:**

The paper addresses the problem of learning on temporal graphs using LLM as backend model.  Authors highlight the recent success of LLMs in in-context learning, reasoning over broad category of tasks. Authors proposes a novel method  TGTalker to adapt LLMs on temporal graphs by converting target link into a LLM prompt by extracting relevant temporal graph like recent history using background set, extract question-answer pairs as few shot examples, query set that formulate link prediction task as natural language and temporal Neighbour sampling mechanism for relevant context. TGTalker also generates natural language explanation.

The authors evaluate their proposed framework against Temporal Graph Neural Networks (TGNN) like TGN, TCNN, GraphMixer and Edge-bank. Authors also explore multiple LLMs like Qwen, Llama, Mistral and GPT4.1 mini.

**Strengths:**

S1. Novel perspective on using pre-trained LLMs to temporal graph domain. This paper explores their capabilities on this domain and answer fundamental questions.
S2. Extensive Evaluation on multiple LLMs
S3. Clear paper writing

**Weaknesses:**

**W1.** A key weakness in the paper is the unclear positioning of its main contribution. There are 2 parallel goals: to show that LLMs can perform temporal link prediction, and that LLMs can also generate textual explanations of those predictions.  The explanations component is largely a by-product of prompting rather than a distinct methodological contribution.  The link-prediction contribution itself requires further clarification. A more focused paper at this early stage of exploring LLMs for temporal graphs would have chosen either to rigorously benchmark LLMs for temporal link prediction or to build and evaluate an LLM-based framework specifically designed for explaining temporal predictions.
**Other weaknesses:**
W2. Lack of competitive TGNN baselines: PINT, TPNet, DyGFormer and CAWN
W3. More clarity is needed on node and time representation: Current framework lacks representation node attributes and it’s unclear how LLMs can relate different nodes with similar neighborhood or attributes. Can it support millions or large-sized graphs? What kind of temporal granularity can be supported?
W3. As shown in the results, high surprise leads to low performance. The key cause seems to be over-dependency on recent temporal neighbours in the prompt context.
W4. Evaluated datasets are rather small in size, with fewer than 10k nodes. This raises questions about the method's utility in real-world large temporal graphs.
W5. Evaluation approach for proposed framework is also not clear. The proposed framework directly outputs the destination node ID. How do you get ranks for other/negative target nodes?
W6. Inference time for TGNN baselines is missing.

The overall claim that LLMs are temporal graph learners is weakly supported; it seems that heuristics encoded in prompts are deriving the majority of the gains.

**Questions:**

Please answer the issues raised in the weakness section above.

Clarification questions:
Q1. Do background sets and example set change with target links?
Q2. Is token vocabulary of backend LLM fixed, or it depends on node ids?

---

> ### Author Response · Authors · 2025-11-24
> **Author Response to Reviewer wyy6 (Part 1)**
>
> We thank the reviewer for the time and consideration of our paper. We would like to address your comments below.
> ---
>
> **W1 unclear positioning**
>
> > A key weakness in the paper is the unclear positioning of its main contribution. There are 2 parallel goals: to show that LLMs can perform temporal link prediction, and that LLMs can also generate textual explanations of those predictions. The explanations component is largely a by-product of prompting rather than a distinct methodological contribution. The link-prediction contribution itself requires further clarification. A more focused paper at this early stage of exploring LLMs for temporal graphs would have chosen either to rigorously benchmark LLMs for temporal link prediction or to build and evaluate an LLM-based framework specifically designed for explaining temporal predictions.
>
> **Response.** Thank you for this high level discussion. Indeed, exploring LLMs for temporal graphs is in its early stage and this work is one of the first to do so. TGTalker has two parallel goals as you mentioned. However, we respectfully disagree that these two goals are necessarily disjoint. When introducing a new family of methods (LLM) into an existing task (temporal link prediction), we believe it is important to demonstrate what significant benefit this new method can bring. Therefore, TGTalker is designed as a framework for both prediction and explanation and shows that LLM can bring a new capability to TG learning outside of what is possible with current TGNNs. We believe TGTalker will serve as the foundation for future work that further expand upon the two goals as you mentioned.
>
> **W2 baselines**
>
>
> > Lack of competitive TGNN baselines: PINT, TPNet, DyGFormer and CAWN
>
> **Response.** Thanks for the discussion. In the paper, we have compared 6 LLM variants in TGTalker against 7 TGNN models across both event-based and snapshot-based models, demonstrating that TGTalker can perform competitively with existing models. We leave additional comparison as future work.
>
> **W3 More clarity**
>
> > More clarity is needed on node and time representation: Current framework lacks representation node attributes and it’s unclear how LLMs can relate different nodes with similar neighborhood or attributes. Can it support millions or large-sized graphs? What kind of temporal granularity can be supported?
>
> **Response.** Thank you for this question.  TGTalker framework is designed to be easily extendable: when node features are textual (e.g., node types, attributes, or descriptions), they can be directly incorporated into the prompt. To demonstrate this, we conducted additional experiments by including additional node feature information specifying which type of node it is in the TGBL-Wiki dataset. The table below shows the impact of adding this feature:
>
> | Model                     | w/o Node Type | w/ Node Type |
> |---------------------------|----------------|----------------|
> | Meta-Llama-3-8B-Instruct  | 0.6036         | **0.6440**     |
> | Qwen3-8B                  | 0.6505         | **0.6512**     |
> | Qwen2.5-7B-Instruct       | **0.6475**         | 0.6471         |
> | Qwen3-1.7B                | **0.6485**         | 0.5811         |
>
> - **how LLM related different nodes**. LLMs can relate different nodes by examining their neighborhood from the background set and the temporal neighbors in the TGTalker framework. In Table 2 LLM explanation categories, we see that in the new node category, the LLM infers that source node 997 likely behaves similarly to node 1546 as their historical interactions are similar. This shows that LLM can relate and identify similar neighborhoods across different nodes
>
> - **support large-sized graphs**. Yes, TGTalker identifies the most salient temporal subgraph as the input context for LLM thus it can extend to large-sized graph. However, the LLM context remains limited by the maximum context size of the base LLM model. With rapid improvement in LLM research, TGTalker can always benefit from further increased context window size of LLMs in the future.
>
> - **temporal granularity**. Thanks, in the paper, we mainly examine continuous time dynamic graphs or CTDGs as they are most common in TGL literature. However, the TGTalker framework is general and you can always specify any timestamp granulairty as the dataset requested.

---

> > ### Author Response · Authors · 2025-11-24
> > **Author Response to Reviewer wyy6 (Part 2)**
> >
> > **W4 recent neighbor dependency**
> >
> > > As shown in the results, high surprise leads to low performance. The key cause seems to be over-dependency on recent temporal neighbours in the prompt context.
> >
> > **Response.** Thank you for this observation. Indeed, high surprise datasets tend to have lower performance as the test edges are more inductive thus harder for the models to predict. It has been shown previously that TGNNs tend to memorize [1] and even under-perform memorization based baseline such as EdgeBank [2]. Because recency bias and repeated behaviors are strong inductive biases in temporal graphs and many TGNNs often utilize these aspects too.
> > However, we observe that larger LLM such as GPT-4.1-mini in the TGTalker framework consistently outperforms the EdgeBank baseline across all datasets in Table 1 thus showing that LLMs learns to reason beyond the recent temporal neighbors. In addition, Table 2 explanation categories showed that LLMs can also recognize alternating patterns and similar node neighborhoods as well.
> >
> > [1] Huang, Shenyang, et al. "Temporal graph benchmark for machine learning on temporal graphs." Advances in Neural Information Processing Systems 36 (2023): 2056-2073.
> >
> > [2] Poursafaei, Farimah, et al. "Towards better evaluation for dynamic link prediction." Advances in Neural Information Processing Systems 35 (2022): 32928-32941.
> >
> >
> > **W5 evaluated dataset size**
> >
> > > Evaluated datasets are rather small in size, with fewer than 10k nodes. This raises questions about the method's utility in real-world large temporal graphs.
> >
> > **Response.** Thank you for this question. In TGTalker, the LLMs are evaluated across 5 datasets to ensure the diversity in evaluation. The TGTalker framework focus on extracting the most salient temporal subgraph as the input context to the LLM and the context subgraph size doesn’t depend on the overall graph size. While scaling to large datasets are possible, it is beyond our current compute capabilities mainly due to the large number of link prediction queries (millions of LLM inference passes). The evaluation in the paper is designed to show how LLM can compare favorably to TGNNs while providing textual explanations.
> >
> > **W6 evaluation approach**
> >
> > > Evaluation approach for proposed framework is also not clear. The proposed framework directly outputs the destination node ID. How do you get ranks for other/negative target nodes?
> >
> > **Response.** Thank you for this question, for which we have clarified the corresponding sections in our paper. We clarify that the LLM directly generates the destination node ID thus it is a generation task. To compare with existing TGNN approaches which are required to output a probability vector ranking all destinations, we use an indicator vector to represent the LLM output, with the predicted destination labelled with 1.0 probability while the rest is set to 0. In this way, the MRR metric can be computed and then TGTalker LLMs can be evaluated fairly with existing TGNN approaches.
> >
> > **W7** inference time
> >
> > > Inference time for TGNN baselines is missing.
> >
> > **Response.** Thank you for this question. Here are details on LLM inference time for TGTalker with Qwen3-8B and Llama3-8B. We report directly the inference time of LLM on the test set for temporal link prediction with an NVIDIA Quadro RTX 8000 GPU (48GB).
> >
> > | Dataset | Model | Inference time (s) | Approx. Hours |
> > |--------|----------|----------| ----------|
> > | tgbl-wiki | Qwen3-8B | 10773 | 3 |
> > | tgbl-wiki | Llama3-8B| 10377 | 3 |
> > | Reddit | Qwen3-8B | 50297 | 14 |
> > | Reddit | Llama3-8B | 45137 | 13 |
> > | LastFM | Qwen3-8B | 86025 | 23 |
> > | LastFM  | Llama3-8B | 84341 | 23 |
> > | UCI | Qwen3-8B | 4046 | 1 |
> > | UCI  | Llama3-8B | 3839 | 1 |
> > | Enron | Qwen3-8B | 7755 | 2 |
> > | Enron | Llama3-8B | 7304 | 2 |
> >
> > We observe that on all datasets, the inference time of LLMs is less than a day. In particular, most datasets like tgbl-wiki, UCI and Enron only have inference time in a few hours.
> >
> > **W8 overall claim**
> >
> > > The overall claim that LLMs are temporal graph learners is weakly supported; it seems that heuristics encoded in prompts are deriving the majority of the gains.
> >
> > **Response.** We respectfully disagree with the reviewer on this point. Table 1 results shows that in the TGTalker framework, LLMs can achieve competitive performance to 7 TGNN baselines across 5 real world datasets and outperforming well-known SOTA baselines such as GraphMixer, TGN and HTGN. TGTalker also outperforms the heuristic baseline EdgeBank. TGTalker explains its predictions in ten distinct categories including alternating sequential patterns to select the globally popular node as destination for unseen source nodes. Therefore, we argue that TGTalker performance strongly supports the claim that LLMs are temporal graph learners.

---

> > > ### Author Response · Authors · 2025-11-24
> > > **Author Response to Reviewer wyy6 (Part 3)**
> > >
> > > **Q1 background and example set**
> > >
> > > > Do background sets and example set change with target links?
> > >
> > > **Response.** Yes, Both the background set and example set are constructed from the most recent edges prior to current query time (changes as time moves forward).
> > >
> > >
> > > **Q2 token vocabulary**
> > >
> > > > Is token vocabulary of backend LLM fixed, or it depends on node ids?
> > >
> > > **Response.** The token vocabulary of the backend LLM is fixed, we use the LLM out-of-the-box so the token vocabulary is the same as the base LLM model.

---

### Official Review · Reviewer_p4VJ · 2025-11-01

**Soundness:** 1
**Presentation:** 3
**Contribution:** 1
**Rating:** 2
**Confidence:** 3

**Summary:**

The paper presents TGTalker, a framework that leverages pre-trained LLMs for learning on temporal graphs. The method reformulates temporal link prediction as a text-based learning problem: the evolving graph is serialized into natural language and provided to an LLM together with recent structural context and examples. Through this prompting strategy, TGTalker enables LLMs to predict future links. Later, the article claims that  TGTalker generates human-readable explanations for its predictions, without any fine-tuning or task-specific training. Experiments across real-world temporal networks show that the approach achieves reasonable performance.

**Strengths:**

Although it does not involve any task-specific training, TGTalker achieves performance comparable to state-of-the-art temporal graph neural networks.

The paper is clearly written and well organized, making it easy to read and understand.

**Weaknesses:**

The paper mainly demonstrates that LLMs can be applied to temporal graphs but offers little theoretical understanding of why this works.
It lacks a principled analysis of temporal reasoning, structure encoding, or generalization, so the contribution feels more empirical than conceptual.

The approach treats LLM prompting as a black box, missing opportunities to relate findings to established principles like message passing, memory models, or temporal inductive bias.

The evaluation lacks rigorous interpretation of what the model learns or how explanations relate to reasoning. The “explainability” component is superficial, as the model’s justifications are unverifiable.

The approach ignores node features, which are often high-dimensional embeddings (e.g., bag-of-words). Directly adding them would further shorten the already limited context window, making scalability worse. This is a major limitation, especially since node features can be essential and may also evolve over time.

**Questions:**

1) How would your framework handle node or edge features, especially when they are high-dimensional or time-varying?
2) Given the token-length limitation of LLMs, how does TGTalker scale to large temporal graphs or long histories? This is particularly important in social domains, where periodicity of interactions arises.
3) How far into the future can your model reasonably predict before recency bias dominates? Did you test for long-range temporal dependencies?
4) How sensitive are the results to prompt phrasing, number of examples (k-shot), or ordering of edges in the background set?
5) Understanding vs. Pattern Matching – How can you be sure that the LLM is performing genuine temporal reasoning rather than exploiting short-term recency or frequency heuristics?
6) Since explanations are generated and classified by the same LLM, how do you avoid circular validation? Have you tested explanation consistency across models or prompts?
7) Have you evaluated TGTalker on datasets with different temporal dynamics (e.g., periodic, bursty, or irregular) to assess robustness?

---

> ### Author Response · Authors · 2025-11-24
> **Author Response to Reviewer p4VJ (Part 1)**
>
> We thank the reviewer for the time and consideration of our paper. We would like to address your comments below.
>
> ----
>
> **W1 understanding of why this works**
>
> > The paper mainly demonstrates that LLMs can be applied to temporal graphs but offers little theoretical understanding of why this works. It lacks a principled analysis of temporal reasoning, structure encoding, or generalization, so the contribution feels more empirical than conceptual.
>
> **Response.** Thank you for this question. We believe that our TGTalker approach is an important first step towards understanding LLM capabilities in modeling both the graph structure and temporal reasoning in temporal graph data. Prior work only focuses on small, toy synthetic graphs while TGTalker is the first work to empirically demonstrate that LLMs can achieve comparable performance to TGNNs in real world tasks thus inspiring further work in this direction.
>
> The design of TGTalker is based on well-established principles in temporal graph learning:
> - **Recency bias is a strong inductive bias.** TGTalker focus on incorporating recent edges related to the prediction as context for LLM
> - **Neighbor information is highly important.** The temporal neighbor sampling component in TGTalker provides important information for LLM reasoning to leverage most relevant past edges for the prediction.
>
> We also want to highlight that TGTalker is the first work to examine explainability for temporal graphs in the lens of LLMs. While TGNN can achieve strong blackbox performance, it is very challenging to extract explanations for their prediction due to the spatial and temporal dependencies on temporal graphs. TGTalker motivates the use of LLMs as an explainer on temporal graphs by showing ten diverse explanation categories and providing a new perspective and direction for temporal graph explainability which is crucial for real world tasks.
>
> **W2 missing opportunities to related findings**
>
> > The approach treats LLM prompting as a black box, missing opportunities to relate findings to established principles like message passing, memory models, or temporal inductive bias.
>
> **Response.** Thanks for the discussion. The goal of this work is to examine the out-of-the-box capability of LLMs on temporal graph tasks. More specifically, we want to examine if LLMs can reason over the complexity of spatial and temporal dependencies on temporal graphs without external help such as 1). semantic information from the text, 2). softprompting techniques that are designed to solve the task. From this perspective, while leveraging TGNNs with message passing and memory modules might be interesting, it is not the focus of this work. In addition, based on the performance comparison with TGN model in Table 1 with TGTalker LLM models, the message passing and memory module in TGN might not even provide additional empirical advantage as it is consistently outperformed by LLMs with text alone.
>
>
> **W3 evaluation**
>
> > The evaluation lacks rigorous interpretation of what the model learns or how explanations relate to reasoning. The “explainability” component is superficial, as the model’s justifications are unverifiable.
>
> **Response.** To validate the correctness of the generated LLM explanations, we conduct a human annotation experiment as seen in Section 5.2 Table 3. We find LLM-generated explanations are rated very correct with a low hallucination rate, and considerable agreement between human-attributed categories and those generated by LLMs.
> We also want to note that there is no ground-truth for the reasoning for future link prediction as it is hard or impossible to obtain such labels (especially for real-world datasets). However, with LLM’s textual explanations, it offers interesting alternative and can be checked for plausibility.
>
>
> **W4 node features**
>
> > The approach ignores node features, which are often high-dimensional embeddings (e.g., bag-of-words). Directly adding them would further shorten the already limited context window, making scalability worse. This is a major limitation, especially since node features can be essential and may also evolve over time.
>
> **Response.** Thank you for this discussion. We agree that incorporating node features has the potential to further enhance the performance of TGTalker. Our framework is designed to be flexible and easily extendable: when node features are textual (e.g., node types, attributes, or descriptions), they can be directly incorporated into the prompt. To demonstrate this, we conducted additional experiments by including additional node feature information specifying which type of node it is in the TGBL-Wiki dataset. The table below shows the impact of adding this feature:
>
> | Model | w/o Node Type | w/ Node Type |
> |----------|-------|----------|
> | Meta-Llama-3-8B-Instruct  | 0.6036| **0.6440**     |
> | Qwen3-8B   | 0.6505  | **0.6512**     |
> | Qwen2.5-7B-Instruct  | **0.6475**  | 0.6471         |
> | Qwen3-1.7B   | **0.6485**  | 0.5811  |

---

> > ### Author Response · Authors · 2025-11-24
> > **Author Response to Reviewer p4VJ (Part 2)**
> >
> > **Q1 features**
> >
> > > How would your framework handle node or edge features, especially when they are high-dimensional or time-varying?
> >
> > **Response.** We would like to emphasize that the focus of this work is to demonstrate that LLM can reason well over temporal graphs and achieve competitive performance to TGNNs while providing the unique advantages of textual explanations for their predictions. TGTalker framework allows features to be incorporated into the LLM context easily, however efficiently incorporating high-dimensional feature vectors can indeed be an interesting future direction. With the promising performance of LLMs shown in this work, we believe there is plenty of opportunities for follow-up work to TGTalker and a new category of LLM based methods for temporal graph reasoning. Lastly, time-varying features are easily handled in TGTalker by attaching them to the edge tuple (src, dst, timestamp, feature).
> >
> >
> > **Q2 token-length**
> >
> > > Given the token-length limitation of LLMs, how does TGTalker scale to large temporal graphs or long histories? This is particularly important in social domains, where periodicity of interactions arises.
> >
> > **Response.**  Thank you for raising this point . We discuss the scalability limitations of our approach in Appendix B in the supplementary material. As with most LLM-based methods, TGTalker’s scalability is primarily constrained by the context window of the underlying language model, rather than by the framework’s ability to process large temporal graphs. Our method can transform temporal graphs of arbitrary size into a textual format; however, only a subset can be accommodated within the model's input window at any given time.
> >
> > That said, rapid advancements in LLM architecture are quickly addressing this limitation. For instance, while the context window of ChatGPT was limited to 4,096 tokens at launch, recent models like GPT-4o now support up to 128,000 tokens—a 30x increase in capacity. TGTalker is well positioned to immediately benefit from such improvements in LLM scalability without requiring architectural changes.
> >
> >
> > **Q3 into the future**
> >
> > > How far into the future can your model reasonably predict before recency bias dominates? Did you test for long-range temporal dependencies?
> >
> > **Response.** Thank you for this interesting question. In current literature on temporal graphs, the link prediction query is always formulated as predicting the next immediate link, i.e. given a source node $u$ and a query timestamp $t$, forecast which destination node this link will reach. The effect of this means that standard evaluation is focusing on short horizon prediction (as seen in most works [1,2,3]). Therefore, in this work, we also compare with TGNNs under this standard short horizon forecast setting. We agree that understanding long-range dependency is an important topic, however how to study it systematically on real-world datasets remains unclear. One can define long-range in both spatial-sense (how many hops away) and temporal-sense (how far in the future can it go). Maybe a recent benchmark T-GRAB[4] would be a good starting point to test this synthetic though it is outside of the scope for this work.
> >
> > [1] Rossi, Emanuele, et al. "Temporal graph networks for deep learning on dynamic graphs." arXiv preprint arXiv:2006.10637 (2020).
> >
> > [2] Huang, Shenyang, et al. "Temporal graph benchmark for machine learning on temporal graphs." Advances in Neural Information Processing Systems 36 (2023): 2056-2073.
> >
> > [3] Poursafaei, Farimah, et al. "Towards better evaluation for dynamic link prediction." Advances in Neural Information Processing Systems 35 (2022): 32928-32941.
> >
> > [4] Dizaji, Alireza, et al. "T-GRAB: A Synthetic Diagnostic Benchmark for Learning on Temporal Graphs." arXiv preprint arXiv:2507.10183 (2025).

---

> > > ### Author Response · Authors · 2025-11-24
> > > **Author Response to Reviewer p4VJ (Part 3)**
> > >
> > > **Q4 sensitivity analysis**
> > >
> > > > How sensitive are the results to prompt phrasing, number of examples (k-shot), or ordering of edges in the background set?
> > >
> > >
> > > **Response.** We refer to the Table below (which is also Table 8 in the paper’s appendix). We can see the results are lightly sensitive to the presence of examples and the background set, but did not explore with changing ordering within them. This is because, regardless of the set they may be attributed to, edges are always temporally ordered to keep temporal coherence, fixing the ordering across components.
> > >
> > > | **Configuration**              | `tgbl-wiki` | Reddit |
> > > |-------------------------------|-------------|--------|
> > > | **TGTalker (All components)**  | 0.649       | 0.613  |
> > > | *w/o In Context Learning*     | 0.645       | 0.612  |
> > > | *w/o Temporal Neighbours*     | 0.322       | 0.122  |
> > > | *w/o Background Set*          | 0.648       | 0.618  |
> > > | *with no components*          | 0.008       | 0.002  |
> > >
> > > We did some preliminary exploration on prompt phrasing, but given the infinite space of possibilities, did not conduct formal ablation studies on that end. As the Table above shows, we also found that temporal neighbours beared a great importance on the model. As explored in the Table below (Table 9 in the paper’s appendix), the number of neighbours directly plays a role, positively correlated with the performance.
> > >
> > >
> > > | **Configuration**                                 | `tgbl-wiki` | Reddit |
> > > |--------------------------------------------------|-------------|--------|
> > > | **TGTalker (nbr = 2)**                             | 0.649       | 0.613  |
> > > | *Number of temporal neighbours = 0*               | 0.322       | 0.122  |
> > > | *Number of temporal neighbours = 1*               | 0.646       | 0.520  |
> > > | *Number of temporal neighbours = 5*               | 0.652       | 0.635  |
> > > | *Number of temporal neighbours = 10*              | 0.656       | 0.643  |
> > >
> > > **Q5 temporal reasoning**
> > >
> > > > Understanding vs. Pattern Matching – How can you be sure that the LLM is performing genuine temporal reasoning rather than exploiting short-term recency or frequency heuristics?
> > >
> > > **Response.** Thank you for this question. As TGTalker LLMs are not fine-tuned on the temporal graph data, any reasoning patterns it discovers are reasoning patterns inherent to the LLM itself. We would like to answer this question in two perspectives:
> > >
> > > - **heuristics on temporal graphs.** We argue that heuristics on temporal graphs are useful and memorization baseline such as EdgeBank have been shown to be highly effective (on datasets with high repetition) [1] and are known to be necessary tools to benchmark TGNNs against. Therefore, discovering heuristics on temporal graphs by LLM is rather natural as it is part of the temporal reasoning toolkit for TG modality.
> > > - **beyond heuristics.** In the 10 explanation categories shown in TGTalker (see Table 2), we observe there are more advanced reasoning pattern beyond recency bias or frequency. For example, the alternation logic pattern shows that LLMs can recognize alternative interactions between nodes such as a source node alternative between two destination nodes in ABAB patterns. These are novel discoveries previously unknown on the TG literature as textual explanations are only made possible in LLMs (until now).
> > >
> > > [1] Huang, Shenyang, et al. "Temporal graph benchmark for machine learning on temporal graphs." Advances in Neural Information Processing Systems 36 (2023): 2056-2073.
> > >
> > >
> > > **Q6 circular validation**
> > >
> > > > Since explanations are generated and classified by the same LLM, how do you avoid circular validation? Have you tested explanation consistency across models or prompts?
> > >
> > >
> > > **Response.** Thank you for this question. When we were designing TGTalker, we find that using chain of thought with Qwen models lead to circular validation and often the models fail to reach an answer. Therefore, in TGTalker, we use in-context examples to guide the model to generate the answer correctly. In terms of explanation correctness, we have conducted the human annotation experiment, see more on W3 and Section 5.2 Table 3.
> > >
> > >
> > > **Q7 dataset robustness**
> > >
> > > > Have you evaluated TGTalker on datasets with different temporal dynamics (e.g., periodic, bursty, or irregular) to assess robustness?
> > >
> > > **Response.** Thanks, yes. In Table 1, we evaluate TGTalker across 5 datasets with diverse patterns (detailed datasets statistics are found in Table 4). LastFM is a bipartite user-song interaction network, where the patterns are more periodic as listening patterns are consistent. On the UCI social network, it is a growing network where new students are constantly registered into the network thus it has the highest surprise index (higher ratio of novel edges). Lastly, tgbl-wiki and Reddit have high repetition patterns thus edges repeat frequently. We observe that across all datasets, TGTalker achieves competitive performance with TGNNs.

---

> > > > ### Comment · Reviewer_p4VJ · 2025-11-27
> > > >
> > > > Thank you for the detailed rebuttal. I appreciate the clarifications and the additional ablation results, which helped address some of my earlier concerns. Based on this, I am willing to raise my score to 4.
> > > >
> > > > That said, several key points remain insufficiently addressed. In particular, the responses regarding theoretical grounding (why and how LLMs actually perform temporal reasoning), scalability under strict context-length constraints, and long-range temporal dependencies were quite high-level and did not offer concrete evidence or methodological insight. These responses were helpful but remained somewhat high-level, and additional substantive clarification would strengthen them.

---

> > > > > ### Author Response · Authors · 2025-11-29
> > > > > **Author Response to Reviewer p4VJ**
> > > > >
> > > > > We thank the reviewer for the constructive discussion throughout the rebuttal period and for raising their score from 2 to 4. We appreciate the feedback regarding theoretical grounding and long-range reasoning, and we agree that these are valuable directions for deeper follow-up work.
> > > > >
> > > > > As highlighted earlier, our goal in this paper is to introduce the first framework that systematically probes how pre-trained LLMs approach temporal graph tasks. Because the models are not finetuned on the data, the reasoning patterns we observe (from known temporal heuristics to more structurally complex behaviors as seen in the additional categories, such as alternation logic) reflect the LLMs’ intrinsic capabilities. We hope the explanation taxonomy and empirical findings help clarify these behaviours and establishes a foundation for future investigation.
> > > > >
> > > > > We again thank the reviewer for the constructive dialogue.

---

### Note · Authors · 2026-01-12

**Comment:**

We, the authors, thank the area chairs for overseeing this fruitful review process. We would particularly like to thank each reviewer for their insightful feedback which we found to be very valuable. We believe our work could benefit from further refinements following the received reviews, and in result of this, withdraw the current paper submission.

**Withdrawal Confirmation:**

I have read and agree with the venue's withdrawal policy on behalf of myself and my co-authors.